# The Scaling Laws of Classification Networks: Insights from Adaptive Exact Average Density Approximation

## Abstract

Our main goal is to establish a generalization bound for classification tasks that aligns with the empirical scaling laws observed in deep neural networks (DNNs). Under the assumption that the boundary of the target classification function is a semi-algebraic set, we show that the generalization error bound can follow scaling laws for large networks. The rate of scaling with respect to sample size is intrinsically linked to the effective dimension of the data manifold, independent of the specific network model or learning algorithm applied. In contrast, the scaling law with respect to the number of parameters varies across learning methods and network architectures. This variability in the parameter scaling law can be quantified by the notion of "box-module dimension, " which measures how the number of model parameters grows as the radius of covering balls decreases, capturing the complexity of the target classification boundary. Using this scaling law, we empirically demonstrate the feasibility of predicting the generalization errors of larger models from those of smaller models.

## 1 Introduction

Scaling laws in deep neural networks (DNNs) indicate that, across a wide range of architectures and tasks, the generalization error is primarily driven by scale. The generalization error typically decreases according to predictable power-law relationships associated with variables such as the number of parameters, the amount of training data, and the computational resources used for training. This observation suggests that larger models, trained on larger datasets, generally perform better in a smooth, quantifiable manner when properly optimized and regularized during training. Such regularities have been instrumental in guiding the design of large-scale models, such as GPT and PaLM, which use scaling curves to forecast performance before full-scale training.

Our approach builds on previous efforts in Bisla et al. (2021) and Sharma and Kaplan (2022) to explain scaling laws, while further highlighting how increasing the depth of DNNs enhances their ability to capture finer local structures, thereby improving predictions. Our analysis relies on two conditions. First, the sample points must densely cover the input domain such that each decision region of a classification network contains at least one training point, and second, the histogram over decision regions of a classification network must accurately reflect the regional average density. By combining these elements, we first express the generalization error in terms of ball radius. We then demonstrate a scaling law under which greater network depths and larger sample sizes lead to a reduced ball radius, thereby bounding the generalization error.

Our contributions are threefold. First, consistent with previous findings of Cortes et al. (1993), Bahri et al. (2024), and Bisla et al. (2021), we demonstrate that the relationship between generalization error and sample size obeys a scaling law, whose slope is inversely proportional to the effective or intrinsic dimension of the data manifold.

Second, we identify another scaling law relating generalization error to network size, particularly as network depth increases. To characterize the rate of this scaling law, we introduce the notion of box-module dimension —a quantity that measures how the number of parameters in a family of models grows as the box size varies

when describing the shape of a set across scales. We show that network architecture and optimization methods can substantially influence this dimension. Moreover, this scaling law enables generalization errors of larger models to be estimated from smaller ones, thereby informing the resources required to increase model depth and the parameter adjustments needed to attain a target level of accuracy.

Third, the prevailing view holds that scaling laws operate within the interpolating regime, where training error is zero, closely linked to the double descent and benign overfitting phenomena observed empirically in deep neural networks (we extend our analysis to this regime in the final appendix). Our findings offer a new perspective: the interpolating regime is not the sole source of scaling laws. Such laws can also emerge when sample points cover the input domain sufficiently densely and the histogram defined over decision regions faithfully approximates the underlying average density.

This document is organized as follows: Section 2 provides an overview of prior research in the field. In Section 3, we establish a bound on the generalization error using specific assumptions about the training points within the geometry induced by a classification network, based on the radius of covering balls. In Section 4, we present scaling laws linking the generalization bound to the number of training points by regularizing the density function. Following this, we introduce the box-module dimension to connect the radii of covering balls to the number of network parameters, thereby establishing scaling laws governing model size. Finally, we show that the generalization error of a large model can be predicted from smaller models using published experimental data. Section 5 offers a mathematical justification for our assumptions, while Section 6 presents conclusions and suggestions for future research.

## 2 Related works

The paper by Hestness et al. (2017) serves as a precursor to the scaling law for accuracy. It demonstrates that large-scale empirical evidence across vision, speech, and language tasks that test loss follows a power-law scaling with dataset size $N$: $aN^{-\alpha} + b$. Rosenfeld et al. (2019; 2021) investigate how the generalization error of neural networks depends on both model size (specifically width and depth) and dataset size. They propose a parametric functional form that integrates model size and data size to predict the generalization error for a range of tasks in vision and language. Hutter (2021) derives power-law learning curves with a generalization error of $\mathcal{O}(1/N^{\beta})$, where $N$ represents the number of training points. The rate $\beta$ is related to the structural properties of the hypothesis space, the regularity of the data, and the complexity of hierarchical models. The Temporal Scaling Law examines test loss evolution in LLMs as training steps increase, using a dynamic hyperbolic model for predictions Xiong et al. (2024). Recent studies improve scaling law reliability through intermediate checkpoints and hyperparameter predictions Choshen et al. (2024) Li et al. (2025).

Some analysis aims to explain the scaling laws. Sharma and Kaplan (2022) introduce a model that explains the emergence of scaling laws based on the number of training points, denoted as $D$, and the number of network parameters, represented as $P$. Bahri et al. (2024) initializes the study of the generalization errors with respect to $D$ and $P$ in the resolution-limited and variance-limited regimes. In the resolution-limited regime, either $D$ or $P$ can approach infinity, allowing the other parameter to scale to infinity. In contrast, the variance-limited regime involves fixing a finite value on one of the parameters while the other can change. This work is compatible with the result by Park et al. (2025), which points out that benign overfitting can occur in the classical regime for large $D$. The study by Bisla et al. (2021) connects test points to the convex hull of their nearest training points within a hypercube, allowing for the approximation of density as uniform, and derives a closed-form expression for generalization error, which can be evaluated numerically and aligns with theoretical results.

From studying tractable shallow networks, it is possible to derive the factors governing the underlying dynamic behaviour at different scales. The foundational work on benign overfitting by Bartlett et al. (2020) closely links to scaling laws and highlights that, for effective generalization, the number of "unimportant" parameter directions must exceed the sample size by a significant margin. Cui et al. (2021) offers a bound on the generalization of kernel ridge regression, showing that eigenvalue decay influences learning rates. Maloney et al. (2022) developed a mathematically tractable joint model with one hidden layer and solved this model in a double limit—considering both large datasets and numerous parameters—they derived analytical formulas

that connect the spectral properties of the data to the power-law scaling of test loss with dataset size or parameter count, explaining why scaling laws can eventually plateau.

# 3 Geometry, sample density, and generalization error

## 3.1 Assumptions and simplifications

We outline simplifications necessary to render our analysis tractable and to focus on the most critical parameters. This analysis focuses on the unit $d$-ball centered at $\mathbf{0}$, denoted as $B_d(\mathbf{0}, 1)$, within an input domain of dimension $d$. This choice does not limit our analysis, as it can be easily adapted to include any compact input domain typically encountered in real-world applications. Additionally, we will not consider the boundary in our analysis because of the compact support, which simplifies and yields more concise results.

Moreover, in this context, we do not differentiate between the dimension of the input domain and the effective dimension, also known as the intrinsic dimension. Given a density function, the effective dimension is the minimum number of independent variables or parameters needed to represent it without substantial loss of information Levina and Bickel (2004); Ansuini et al. (2019). This effective dimension is often significantly smaller than the ambient dimension of the input domain, which mitigates the Curse of Dimensionality: while the sample complexity grows exponentially with the ambient dimension $d$, replacing $d$ with the effective dimension $d_e \ll d$ yields a substantially more favorable scaling. Note that our results derived with $d$, the dimension of the input domain, can be replaced with $d_e$, the effective dimension of the density function $\mathcal{P}$. Additionally, we sometimes use $d_e$ instead of $d$ to align with concepts discussed in previous works.

The generalization error measures the difference between the expected test error and the expected training error, while the generalization bound represents an upper bound of that error. Our analysis makes two assumptions about training data distribution.

**Assumption of densely covering**: We assume that the data is sufficiently densely distributed, allowing spheres centered at a selected subset to cover the input domain within a defined radius.

To access the density of training data in the input domain, we use the $\beta_l$-covering of the input domain, which means that for any $\mathbf{x}$ in the input domain, there is a training point $\mathbf{x}_i$ such as $\|\mathbf{x} - \mathbf{x}_i\|_2 \leq \beta_l$. The $\beta_l$-covering distribution in training points emphasizes that the input domain can be covered using balls with radius $\beta_l$, and each ball contains at least one training point.

The classification network $\mathcal{M}$ is responsible for learning the classifier function $f$ by utilizing $N$ training data pairs $(\mathbf{x}_i, f(\mathbf{x}_i))$. $\mathcal{M}$ is a piecewise constant function, dividing the input domain into non-overlapping decision regions and assigning each region a specific class.

**Assumption of histogram approximation**: We assume that sufficient training data are available for each decision region to allow an accurate approximation of the average probability mass within the region using a histogram over bins defined by the regions. Note that this assumption concerns the average density function within regions, not the actual density function.

These two assumptions form the basis of our analysis on generalization bounds of classification functions, which help us identify the geometric parameters contributing to predicting errors in unseen data. In Section 5, we will provide an analysis that sufficiently supports these assumptions.

## 3.2 Generalization bound for covering with fixed radius balls

To understand the local geometry of the input of a neural network's function, we use covering balls in the input domain, focusing on two parameters: $\gamma_s$, the radius of the smallest enclosing ball for a set of regions, and $\beta_l$, the radius of the largest inscribed ball within that set. These regions relate to the arrangement of training points in the decision areas in classification networks. In Figure 1(a), $\gamma_s$ is the radius of the ball encompassing both regions, while $\beta_l$ represents the largest inscribed ball.

Our study utilizes specified balls to analyze local geometry, linking this analysis to generalization error and sample complexity. We demonstrate that the parameter $\gamma_s$ effectively characterizes generalization errors.

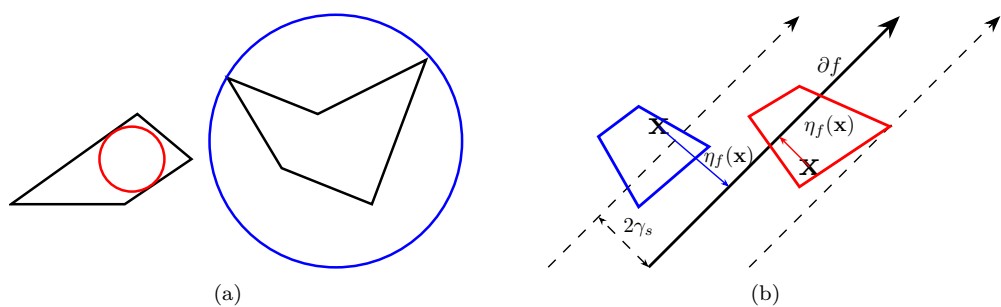

(a)                                                      (b)

Figure 1: (a) Each polygon can be enclosed by and inscribed in a ball. The left (red) ball of radius $\beta_l$ can be inscribed within both polygons, while the right (blue) ball of radius $\gamma_s$ can enclose them. (b) Polygons represent decision regions of a classification network, each assigned a specific class. Any point $\mathbf{x}$ within the $2\gamma_s$-tube of the decision boundary $\partial f$ (i.e., $\eta_f(\mathbf{x}) \le 2\gamma_s$) can potentially cause a classification error, establishing an upper bound on generalization error. The red polygon intersects $\partial f$, with its two subregions representing different classes with respect to $f$, but they belong to the same class in the network. Conversely, any point farther than $2\gamma_s$ from $\partial f$ cannot belong to a polygon that intersects the boundary, like the blue polygon.

However, a higher sample density is essential to ensure that each decision region contains at least one training point. This necessary density is determined by the maximum radius of the balls, $\beta_l$, within the decision regions.

The lemma below shows that the sample density defined by $\beta_l$ ensures that each region formed by the classification network has at least one training point.

**Lemma 1.** *Let $\beta_l$ be the radius of the largest inscribed balls in the finest decision regions of the one-hot classification network $\mathcal{M}$. If samples cover the input domain with a $\beta_l$-covering, then each region of $\mathcal{M}$ contains at least one training point.* ■

**Proof**. Given that $\beta_l$ is the radius of the largest inscribed ball within all decision regions in the input domain derived by network $\mathcal{M}$, it follows that for any region, there exists a ball with a radius of $\beta_l$ contained within that region.

To reach a contradiction, let's assume there exists a region containing no training points. In this case, the center of the inscribed ball with a radius of $\beta_l$ in that region would be devoid of training points within a distance of $\beta_l$. This situation contradicts the assumption that the input domain is $\beta_l$-covering.     Q.E.D.

The derivation of Lemma 2 is detailed in Appendix A. The lemma shows that, for sufficiently large networks, the generalization error in classification tasks exhibits a linear behavior with respect to $\gamma_s$. The analysis is applicable to classification networks that define piecewise-constant functions over the input domain, with at least one training point assigned to each decision region of a specific class.

Let $f$ be the target classification function. For generalization error, we use the volume of enclosing balls near the actual classification boundary, $\partial f$, based on the observation that when $\partial f$ intersects with the network's decision regions, potential errors arise. Specifically, points within $2\gamma_s$ of $\partial f$ are likely to belong to a region that could contribute to errors. This relationship emphasizes $\gamma_s$'s significance in determining the network's generalization error, as shown in Figure 1(b). In addition, to measure the volume of balls proximity to $\partial f$, denoted by $2\gamma_s$, we need to characterize the boundary over the input domain. Here, we assume $\partial f$ is a semi-algebraic set because a semi-algebraic set can realize an arbitrarily complicated shape.

The process of stratification involves decomposing a semi-algebraic set into a finite union of disjoint semi-algebraic subsets, known as strata, which are locally closed submanifolds (Benedetti et al., 1991). In a stratification, the boundary of the entire set $S$ is the union of all strata of lower dimension than the "top-

level" interior strata. If $S$ is a closed semi-algebraic set, then

$$\partial S = \bigcup_{X_j \cap \text{int}(S) = \emptyset} X_j \tag{1}$$

where $X_j$ is a stratum of $S$ and it does not touch the interior of $S$.

The target function $f$ partitions the input domain into a finite union of disjoint sets, denoted as $\bigcup g(i)$ where $i \in [l]$. The set $g(i) = \{\mathbf{x} \in B_d(\mathbf{0}, 1) | f(\mathbf{x}) = \mathbf{e}_i\}$ corresponds to class $i$ and is assumed to be a semi-algebraic set, meaning that its representation is a finite union of disjoint subsets, each defined by a finite combination of polynomial equations and inequalities. Formally, we can represent a stratification of $\bigcup g(i)$ by a finite collection of semi-algebraic subsets $\{f_k = \bigcup_j f_k^j\}$, where the dimension of $f_k^j$ is $k$ and $f_k^j$ are disjoint semi-algebraic sets. We refer to the volume of $f_k^j$ as $|f_k^j|$. The concept of volume measures the size of an object: a 1-volume corresponds to length, while a 2-volume corresponds to area, and so on.

Hence, considering the boundary of the involved sets leads to the following relationship:

$$\partial f = \bigcup_{i=1}^{l} \partial g(i) = \bigcup_{k=0}^{d-1} f_k = \bigcup_{k=0}^{d-1} \bigcup_j f_k^j, \tag{2}$$

with $f_k = \bigcup_j f_k^j$ and

$$|f_k| = \sum_j |f_k^j|. \tag{3}$$

For example, consider the stratification of $g(1) = \{(x, y) \in \mathbb{R}^2 \mid |x| + |y| < 1\} \bigcup \{(x, y) \in \mathbb{R}^2 \mid |x| + |y| = 1\}$ with dimension 2. The first subset of that stratification is the interior of $g(1)$; therefore, that set does not contribute to the boundary of $g(1)$. We can decompose the second subset into four 1-dimensional semi-algebraic sets: $f_1^1 = \{x + y = 1 | x, y > 0\}$, $f_1^2 = \{-x + y = 1 | x < 0, y > 0\}$, $f_1^3 = \{-x - y = 1 | x, y < 0\}$, and $f_1^4 = \{x - y = 1 | x > 0, y < 0\}$, and four 0-dimensional vertices: $f_0^1 = \{(1, 0)\}$, $f_0^2 = \{(0, 1)\}$, $f_0^3 = \{(-1, 0)\}$, and $f_0^4 = \{(0, -1)\}$. The boundaries of their stratification of $g(1)$ are as follows:

$$\partial g(1) = f_1 \cup f_0 = (\cup_i f_1^i) \cup (\cup_j f_0^j). \tag{4}$$

The following lemma demonstrates that assuming the decision boundary is a semi-algebraic set leads to its decomposition into a combination of polynomial functions of varying degrees. This lemma connects the generalization error to the network's granularity, represented by $\gamma_s$ and $\beta_l$, in approximating the classification boundary. For networks with a small $\gamma_s$, the decision boundary that intersects a given region can be considered a curved $(d-1)$-dimensional surface, homeomorphic to a hyperplane. The $2\gamma_s$-neighborhood of the union of the "hyperplanes" represents the region in the input domain that can contribute to a bound of generalization error.

**Lemma 2.** *Let $f$ be the function $B_d(\mathbf{0}, 1) \to \{\mathbf{e}_1, \cdots, \mathbf{e}_l\}$, defined as $f = \bigcup_{i=1}^{l} g(i)$, where $g(i) = \{\mathbf{x} | f(\mathbf{x}) = \mathbf{e}_i\}$ is a semi-algebraic set with decision boundary $\partial f$ adhering to the stratification equations (2) and (3). Let $\gamma_s$ and $\beta_l$ represent the smallest enclosing radius and largest inscribed radius of the decision regions in $B_d(\mathbf{0}, 1)$ of the one-hot classification network $\mathcal{M}$. Assume the training data satisfies the conditions for $\beta_l$-covering the unit $d$-ball and the histogram approximation. We use the notation $|S|$ to denote the volume of the set $S$.*
*(i) For an arbitrary density function $\mathcal{P}$ on $B_d(\mathbf{0}, 1)$, the error is given by:*

$$\int_{B_d(\mathbf{0}, 1)} g(f(\mathbf{x}), \mathcal{M}(\mathbf{x})) \mathcal{P}(\mathbf{x}) \, d\mathbf{x} \le \frac{1}{N} \sum_{i=1}^{N} g(f(\mathbf{x}_i), \mathcal{M}(\mathbf{x}_i))$$

$$+ c_{\mathcal{P}} \sum_{k=0}^{d-1} (2\gamma_s)^{d-k} |B_{d-k}(\mathbf{0}, 1)| |f_k|, \tag{5}$$

where $g(f(\mathbf{x}), \mathcal{M}(\mathbf{x}))$ is the 0/1 error, and $c_{\mathcal{P}} \in (0, 1)$ depends on the density supported over the $2\gamma_s$-neighborhood of $\partial f$.

*(ii) We further suppose the volume of $f_{d-1}$ is not zero. For small value of $\gamma_s$, we can approximate the above expression as follows:*

$$\int_{B_d(\mathbf{0},1)} g(f(\mathbf{x}), \mathcal{M}(\mathbf{x})) \mathcal{P}(\mathbf{x}) \, d\mathbf{x} \leq \frac{1}{N} \sum_{i=1}^{N} g(f(\mathbf{x}_i), \mathcal{M}(\mathbf{x}_i)) + c_{\mathcal{P}}(4\gamma_s)|f_{d-1}|. \tag{6}$$

■

**Remark.** While our analysis focuses on the generalization bounds from the perspective of histogram approximation of the average density in the decision regions of a classification network, all analytical conclusions remain applicable — with slight modifications to the derivations of Lemma 2 — to the interpolating zone of a network in which the training error is zero. Although the interpolating-zone approach yields similar results to Theorems 1 and 2, we relegate it to Appendix F to keep the exposition focused on the histogram-approximation approach. Remarkably, both approaches yield similar results, revealing a new insight: the scaling-law behavior can be achieved via two distinct pathways — either by approximating the average density across all decision regions or by training a network to the interpolating zone.

## 4 Connecting geometry to scaling laws

### 4.1 Scaling laws with respect to sample size

The early work on learning curve theory by Cortes et al. (1993) illustrates how a binary classification network's performance changes as the amount of training data increases. This study reveals a scaling law related to data size: the log-log plot of generalization error versus the number of training samples, with nonzero training error, forms a straight line with a negative slope. For the MNIST dataset, the absolute value of slope ranges from $\frac{1}{2}$ to 1 (see Figures 5 and 6 of the paper).

The conclusions drawn from Lemma 2 support these findings with a regulatory assumption on the density function $\mathcal{P}$. By imposing $\mathcal{P} \geq k > 0$ over $B_{d_e}(\mathbf{0}, 1)$ where $d_e$ is the effective dimension of $\mathcal{P}$, the probability mass of $\mathcal{P}$ over the ball $B_{d_e}(\mathbf{a}, \beta_l)$, denoted as $P(\mathbf{a}, B_{d_e}(\mathbf{a}, \beta_l))$, can be uniformly bounded below as follows:

$$P(\mathbf{a}, B_{d_e}(\mathbf{a}, \beta_l)) = \int_{B_{d_e}(\mathbf{a}, \beta_l)} \mathcal{P}(\mathbf{x}) \, d\mathbf{x} \geq |B_{d_e}(\mathbf{0}, \beta_l)| \cdot \min_{\mathbf{x} \in B_{d_e}(\mathbf{a}, \beta_l)} \mathcal{P}(\mathbf{x}) \geq |B_{d_e}(\mathbf{0}, \beta_l)|k = c\beta_l^{d_e}, \tag{7}$$

where $c = |B_{d_e}(\mathbf{0}, 1)|k$.

Note that, in this context, we do not differentiate between effective and ambient dimensions. Therefore, we substitute the input domain dimension with the effective dimension in Lemma 2. The findings presented below indicating that the generalization error bound for a large network and sample size follow a scaling law with a slope of $1/d_e$.

**Theorem 1.** *The assumptions related to Lemma 2(ii) hold. Suppose the density function $\mathcal{P} \geq k > 0$. The generalization error with respect to sample size for a one-hot classification $\mathcal{M}$ is bounded by $\frac{|\partial f_d|}{(cN)^{1/d_e}}$.* ■

**Proof.** Let $N$ be the number of training points that forms a $\beta_l$-covering. Then, the expected number of training points within a distance $\beta_l$ at $\mathbf{x}$ is given by

$$\sum_{k=1}^{N} k \binom{N}{k} P(\mathbf{x}, B_{d_e}(\mathbf{x}, \beta_l))^k (1 - P(\mathbf{x}, B_{d_e}(\mathbf{x}, \beta_l)))^{N-k} = NP(\mathbf{x}, B_{d_e}(\mathbf{x}, \beta_l)) \geq 1 \tag{8}$$

By the assumption $\mathcal{P} \geq k > 0$ and Eq. (7), there exists a point $\mathbf{x}_0$ and integer $t$ such that $NP(\mathbf{x}_0, B_{d_e}(\mathbf{x}_0, \beta_l)) = t \geq cN\beta_l^{d_e}$. Taking $t = 1$ as the minimal case, we derive the inequality $\frac{1}{(cN)^{1/d_e}} \geq \beta_l$. Applying this inequality, $|f_{d-1}| \leq |\partial f_d|$, and $K\beta_l \geq \gamma_s$ for some $K \geq 1$ (since $\gamma_s$ and $\beta_l$ are both bounded

and bounded away from zero, their ratio $\gamma_s/\beta_l$ is bounded above by some fixed constant $K \geq 1$) to Lemma 2(ii) yields the generalization error bound $c_\mathcal{P}(4\gamma_s)|f_{d-1}| \leq c_\mathcal{P}(4K\beta_l)|f_{d-1}| \leq \frac{4Kc_\mathcal{P}|f_{d-1}|}{(cN)^{1/d_e}} \leq \frac{4Kc_\mathcal{P}|\partial f_d|}{(cN)^{1/d_e}}$. This establishes the scaling law for the generalization error bound of a sufficiently large one-hot network: the bound scales as $\frac{|\partial f_d|}{(cN)^{1/d_e}}$ with respect to sample size. Q.E.D.

### 4.2 Box-module dimension and scaling laws with respect to network size

We have established a connection between the generalization bound and the radii of balls that represent the local geometry of a network function. Our next objective is to investigate how these radii relate to the number of parameters in a neural network.

#### 4.2.1 Box-module dimension

The box-counting dimension is a way to measure the "ruggedness" or complexity of a shape by examining how it occupies space at different scales. To calculate the box-counting dimension of a specific shape, we examine the relationship between the scaling factor and the number of boxes needed to cover it. The dimension $D$ is the logarithmic power law relationship between the number of boxes and their size:

$$D = \lim_{s \to 0} \frac{\log \#B(s)}{\log(1/s)}, \tag{9}$$

where $\#B$ is the number of boxes (a grid of squares) with side length $s$ over the object. This dimension is found by overlaying a grid of boxes with side length $s$ over the object, counting the number of boxes, then making the boxes smaller and counting again, and observing how the count increases as $s$ decreases. In practice, since the box-counting dimension may not exist or may be numerically difficult to calculate, the upper box-counting dimension is used to indicate that this set has a dimension at least $\bar{D}$ by considering

$$\bar{D} = \limsup_{s \to 0} \frac{\log \#B(s)}{\log(1/s)}. \tag{10}$$

The box-counting dimension measures the complexity of a set by quantifying how many boxes are needed to describe its shape at various scales of box size. In contrast, the box-module dimension measures the number of parameters $\#W$ in a model $\mathcal{M}$ required to construct the box-coverage of a target shape when one zooms into the shape. We define the box-module dimension as follows:

$$D_\mathcal{M} = \lim_{s \to 0} \frac{\log \#W(s)}{\log(1/s)}. \tag{11}$$

By this definition, we focus on how the radii decrease with depth as the number of parameters in a neural network increases. We express the relationship between the number of parameters $\#W$ in network $\mathcal{M}$ and the radius $\gamma_s$ as follows:

$$\gamma_s \leq \frac{1}{\#W^{1/\log_2(1+\alpha(\gamma_s))}}. \tag{12}$$

In our analysis of classification tasks, the radius $\gamma_s$ is derived from the decision regions of network $\mathcal{M}$. Increasing the number of network parameters $\#W$ generates more decision regions, potentially reducing the radius. As a result, $\gamma_s$ is a non-increasing function of $\#W$. The condition in Eq. (12) states that the growth rate of $\#W^{1/\log_2(1+\alpha)}$ must be slower than the decrease in $\gamma_s(\#W)$. This condition provides an upper bound of parameters, preventing the network from overfitting or over-approximating the training points. These limitations depend on the network model and training procedure, as characterized by the value of $\alpha(\gamma_s)$.

Taking the logarithm on both sides and then dividing by $\log_2 1/\gamma_s$, we obtain:

$$\frac{\log_2 \#W}{\log_2 1/\gamma_s} \leq \log_2(1 + \alpha(\gamma_s)). \tag{13}$$

As $\gamma_s \to 0$, we obtain the upper box-module dimension:

$$\limsup_{\gamma_s \to 0} \frac{\log_2 \#W}{\log_2 1/\gamma_s} = \bar{D}_{\mathcal{M}} \to \log_2(1 + \alpha_{\mathcal{M}}). \tag{14}$$

Here, the parameter $\alpha_{\mathcal{M}} > 0$ is model-dependent and influenced by factors such as the network architecture and the specific training method used. Due to this model dependence, networks with the same input domain can produce different box-module dimensions depending on their architecture and training method. In Appendix B, we demonstrate that the exact configuration of the input domain, generated by shallow and deep network structures, can produce different box-module dimensions in the scaling laws.

The logarithmic function in Eq. (12) emerged from our attempts to construct a network $\mathcal{M}_0$ and expand its depth to create $\mathcal{M}_1$. The goal was to ensure that $\gamma_1$ of $\mathcal{M}_1$ is reduced to less than half the value of $\gamma_0$ of $\mathcal{M}_0$, while also limiting the increase in the number of parameters in $\mathcal{M}_1$ to less than $1 + \alpha$ times that in $\mathcal{M}_0$. Next, we apply the process recursively to decrease the enclosing radius to a desired value, $\gamma_s$. To reduce the radius from $\gamma_0 = 1$ to $\gamma_s$, we apply the procedure $L = -\log_2 \gamma_s$ times. Let the resulting network be $\mathcal{M}_L$. Using $\#W_{l_k} \leq (1 + \alpha)\#W_{l_{k-1}}$, with initial condition $\#W_0 = 1$, we can deduce the number of parameters in network $\mathcal{M}_L$ with $\#W \leq (1 + \alpha)^L$. Hence, $\log_2 \#W \leq -\log_2(1 + \alpha) \log_2 \gamma_s$. This gives a relation between $\gamma_s$ and $\#W^{\frac{1}{\log_2(1+\alpha)}}$ with $\alpha > 0$ as depicted in Eq. (12). We note that using the same value of $\alpha$ across all sub-networks and depths is not a requirement in this construction. We can easily adapt the design to accommodate different values of $\alpha$ in each sub-network.

We consider the network $\mathcal{M}_L^0 = [I + \rho M_L] \circ [I + \rho M_{L-1}] \circ \cdots \circ [I + \rho M_1]$, where $I$ denotes the identity mapping (serving as a skip connection within each layer), $\rho$ denotes the Rectified Linear Unit (ReLU) activation function, and $M_l$ represents an affine linear mapping. We show that it is possible to arrange the hyperplanes and the training points such that the scaling law is satisfied with a constant exponent for any depth $l$ in $\mathcal{M}_l^0$. The arrangement of the hyperplanes and training points required here is not generally achievable by gradient-based training methods. Furthermore, our construction halves the covering radius at each successive layer. Since the number of covering balls scales exponentially with each such reduction, this halving condition leads to an exponential increase in network width from one layer to the next. This contrasts with practical architectures such as ResNets and VGGs, where network width grows only moderately (e.g., polynomially) across layers. Therefore, the exponential parameter requirement is a property of our theoretical construction, introduced to establish the existence of scaling laws, and should not be interpreted as a constraint on practical architectures. A detailed derivation of this proposition can be found in Appendix C.

**Proposition 1.** *Let $\#W_l$ denote the number of parameters and $\gamma_l$ denote the radius of the smallest enclosing balls of the regions induced by $\mathcal{P}_l^0$ in network $\mathcal{M}_l^0$. There exists a configuration of hyperplanes in layer $\rho M_{l+1}$ and a choice of training points such that the finest partition $\mathcal{P}_{l+1}^0$ of network $\mathcal{M}_{l+1}^0$ with $l \leq L - 1$ is a refinement of $\mathcal{P}_l^0$ that satisfies $\gamma_{l+1} \leq \frac{\gamma_l}{2}$. Additionally, the number of parameters in $\mathcal{M}_{l+1}$ and $\mathcal{M}_l$ can be constrained by the inequality $\#W_{l+1} \leq (1 + \alpha)\#W_l$, where $\alpha \geq 1$ when $l$ is large (as the covering radius becomes smaller).* ∎

The construction in this proposition focuses on how to achieve the desired covering radius by increasing network width at each layer. Once the desired covering radius of $\mathcal{M}_L^0$ is achieved, we align the output dimension with that of the target network by appending a series of layers of gradually decreasing width. This width-decreasing composition preserves the covering radius achieved by $\mathcal{M}_L^0$.

### 4.2.2 Scaling laws with respect to network size and experiments

Now, let's examine the implications of condition on Eq. (12) on generalization bounds. By substituting that condition into Lemma 2 for $\gamma_s$, the generalization bounds become inversely proportional to the number of parameters in a network.

**Theorem 2.** *Suppose the classification boundary $\partial f$ of target classification function $f : B_d(\mathbf{0}, 1) \to \{\mathbf{e}_1, \cdots, \mathbf{e}_l\}$ is a semi-algebraic set adhering to the stratification equations (2) and (3). Let $\gamma_s$ and $\beta_l$ denote the smallest enclosing radius and the largest inscribed radius of input domain partition regions derived by*

one-hot classification network $\mathcal{M} : B_d(\mathbf{0}, 1) \to \{\mathbf{e}_1, \cdots, \mathbf{e}_l\}$. We assume that the training data meets the assumptions of $\beta_l$-covering the unit d-ball and the histogram approximation. Let $\#W_{\mathcal{M}}$ represent the number of parameters in $\mathcal{M}$. Assume that $\gamma_s \leq \frac{1}{\#W_{\mathcal{M}}^{1/\log_2(1+\alpha_{\mathcal{M}})}}$ for some $\alpha_{\mathcal{M}} > 0$, which depends on the learning method used to obtain $\mathcal{M}$ from $N$ i.i.d. training data points $\{(\mathbf{x}_i, f(\mathbf{x}_i))\}$ from $\mathcal{P}$. Let $|S|$ denote the volume of the set $S$.

*(i) We can bound the error between classification function $f$ and $\mathcal{M}$ as follows:*

$$\int_{B_d(\mathbf{0},1)} g(f(\mathbf{x}), \mathcal{M}(\mathbf{x}))\mathcal{P}(\mathbf{x}) \, d\mathbf{x} \leq \frac{1}{N} \sum_{i=1}^{N} g(f(\mathbf{x}_i), \mathcal{M}(\mathbf{x}_i))$$
$$+ c_{\mathcal{P}} \sum_{k=0}^{d-1} \left( \frac{2}{\#W_{\mathcal{M}}^{1/\log_2(1+\alpha_{\mathcal{M}})}} \right)^{d-k} |B_{d-k}(\mathbf{0}, 1)| |f_k|, \tag{15}$$

*where $g(f(\mathbf{x}), \mathcal{M}(\mathbf{x}))$ is the 0/1 error and $c_{\mathcal{P}} \in (0, 1)$ depends on the density supported over the $2\gamma_s$-neighborhood of $\partial f$.*

*(ii) We further suppose the volume of $f_{d-1}$ is not zero. For small value of $\gamma_s$, we can approximate the above expression as follows:*

$$\int_{B_d(\mathbf{0},1)} g(f(\mathbf{x}), \mathcal{M}(\mathbf{x}))\mathcal{P}(\mathbf{x}) \, d\mathbf{x} \leq \frac{1}{N} \sum_{i=1}^{N} g(f(\mathbf{x}_i), \mathcal{M}(\mathbf{x}_i)) + \frac{4c_{\mathcal{P}}|f_{d-1}|}{\#W_{\mathcal{M}}^{1/\log_2(1+\alpha_{\mathcal{M}})}}. \tag{16}$$

∎

Estimating the exact bound presents particular challenges. However, in the log-log plots of the generalization bounds in Eq. (16) against the number of parameters, the slope is $-1/\log_2(1 + \alpha_{\mathcal{M}})$. This slope serves as an index of how efficiently a network family performs as its parameter count increases while reducing generalization error. In the following subsection, we use the slope derived from smaller models to predict the generalization error of larger models.

### 4.2.3 Empirical validation

If we approximate the generalization error with the bound, we can use the following formulation to estimate $\alpha_{\mathcal{M}}$ from smaller models with known generalization errors and numbers of parameters:

$$\left( \frac{ge_1}{ge_2} \right)^{\log_2(1+\alpha_{\mathcal{M}})} \approx \frac{\#W_2}{\#W_1}, \tag{17}$$

where $ge_i$ and $\#W_i$ represent the generalization error and the number of parameters for model $i$, respectively. This approximation does not necessarily require operating in the interpolation zone, where the training error is zero or nearly zero. However, within the interpolation zone, the generalization error can be approximated by the expected test error. Let $\hat{\alpha}$ represent the estimated value of $\alpha_{\mathcal{M}}$ from the two smallest models. We evaluate the prediction accuracy with two prediction methods: Method A uses this estimated value to predict the generalization error $\hat{ge}_{i+1}$ for model $i+1$ using model $i$: $\left( \frac{ge_i}{\hat{ge}_{i+1}} \right)^{\log_2(1+\hat{\alpha})} \approx \frac{\#W_{i+1}}{\#W_i}$, and Method B uses this estimated value to predict the generalization error $\hat{ge}_{i+1}$ with a fixed reference model (model 2): $\left( \frac{ge_2}{\hat{ge}_{i+1}} \right)^{\log_2(1+\hat{\alpha})} \approx \frac{\#W_{i+1}}{\#W_2}$. To evaluate the accuracy of this prediction, in both methods, we calculate the relative error (RE) using the formula: $\frac{|\hat{ge}_{i+1} - ge_{i+1}|}{ge_{i+1}} \times 100\%$.

The proposed methods, which use the two smallest models to estimate the box-module dimension, can be overly optimistic, as the prediction may rely too heavily on the selected small models. The robustness and applicability of this estimation and prediction approach in real-world settings require further investigation.

We evaluate the extrapolation performance of this approach on large models with numerous training points, which is more likely to align with our assumptions and less likely to impose limitations on the depth increment

required in prediction. We conduct two sets of experiments: one for classification tasks (reported in I-III) and the other for a regression task (reported in IV).

For the first set, we illustrate these scaling laws using experimental results from ResNet and VGG models on the ImageNet and CIFAR-100 datasets, drawn from various benchmark implementations, related papers, and reports. The learning algorithms discussed in those documents were carefully implemented, consistently achieving near-zero training errors. This enables us to assume that the expected test error closely approximates the generalization error, and to present our findings in a form directly relevant to researchers and practitioners in the field.

The absolute value of the slope measures the rate of change in test errors with respect to the number of model parameters on the log-log scale. In our analysis of VGGs and ResNets on ImageNet, we found that the absolute value of the slope for VGGs $\approx 1.5$ is larger than that for ResNets $\approx 0.2$. This result suggests that VGGs achieve approximately a 7-fold greater reduction in test error per unit increase in parameters compared to ResNets on a log-log scale. Nevertheless, this efficiency comes at the expense of increased parameter redundancy, as the VGG model has over 100M parameters.

**I. ResNets on ImageNet:**

ImageNet is a vast visual database containing over 14 million images across more than 20,000 categories, aimed at advancing research in object recognition. Table 1 and Figure 2 illustrate the predicted Top-1 Error for the ResNet model on this dataset. As shown in Figure 2, the scaling law indicates that the generalization error bound, which is the difference between the expected test error and the expected training error, behaves as a line in the log-log plot of generalization error and the number of parameters. Compared to Eq. (16), the generalization error bound is inversely proportional to the number of parameters with exponent $\log_2(1+\alpha_{\mathcal{M}})$. In the log-log plane of generalization error versus the number of parameters, this formula becomes a line with slope $-1/\log_2(1+\alpha_{\mathcal{M}})$. The estimated value of $\alpha$ from ResNet-18 and ResNet-34 is 35.14. This results in $\log_2(1+\hat{\alpha}) = 5.188$ and a slope of $-\frac{1}{\log_2(1+\hat{\alpha})} = -0.19$. This slope suggests, according to Equation (17), that for the test error of a ResNet to be reduced to half its original value, the number of parameters would have to increase to thirty-two times its original size. We use this slope value to derive the predicted Top-1 error presented in the table and figure.

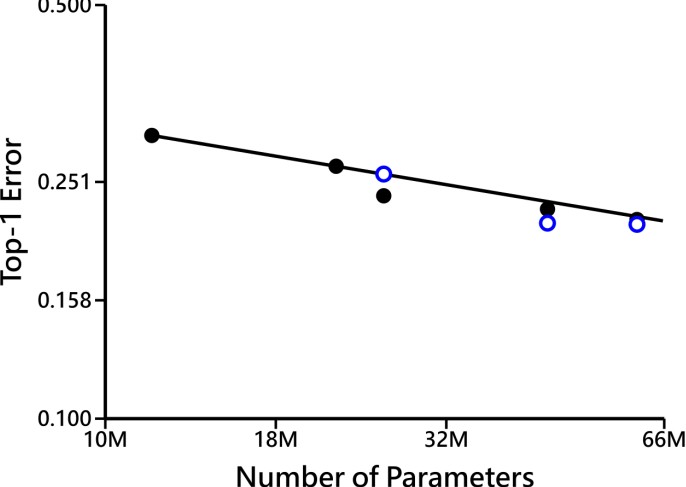

Figure 2: Log-log plot of the data presented in Table 1. The x-axis labels represent the number of parameters, while the y-axis labels indicate the top-1 test error, both displayed in raw values rather than logarithms for ease of interpretation. The line passing through the two leftmost solid points, generated by the smaller models (ResNet-18, ResNet-34), was used to predict the performance of the larger models (ResNet-50, ResNet-101, ResNet-152). Actual data points are shown as solid circles, while predicted points from Method A are shown as empty circles.

| Model-Depth | Size (M) | Error (%) | A: Pred. Err. (%) | A: RE (%) | B: Pred. Err. | B: RE |
|---|---|---|---|---|---|---|
| ResNet-18 | 11.7 | 30.1 | x | x | x | x |
| ResNet-34 | 21.8 | 26.7 | x | x | x | x |
| ResNet-50 | 25.6 | 23.8 | 25.9 | 8.82 | 25.9 | 8.82 |
| ResNet-101 | 44.5 | 22.6 | 21.4 | 5.31 | 23.3 | 3.1 |
| ResNet-152 | 60.2 | 21.7 | 21.3 | 1.84 | 21.9 | 0.88 |

Table 1: Prediction of the top-1 test error of ResNet models on the ImageNet dataset, whose validation set contains approximately 6% label errors Northcutt et al. (2021). The number of parameters and top-1 test errors were obtained from the original paper by He et al. (2016) as well as various benchmark re-implementations Wightman et al. (2021), including those in PyTorch and TensorFlow PyTorch Development Team (2025). There may be slight variations depending on the training setup, such as data augmentation techniques and learning rate schedules. The results predicted from smaller models achieve a relative error of less than 10%, enabling reliable assessment of the resources required for larger models.

## II. ResNets on CIFAR-100:

The CIFAR-100 dataset consists of 60,000 color images, each measuring 32x32 pixels, and it includes 100 different classes, with 600 images per class. The dataset is divided into 50,000 training images and 10,000 test images. The Top-1 Error, as shown in Table 2, was obtained from the implementation of ResNet models on the CIFAR-100 dataset, as reported by Jafar and Lee (2023). The slope derived from the ResNet-20 and ResNet-34 models is -0.15, with an estimated value of $\alpha$ being 93.2.

| Model-Depth | Size (M) | Error (%) | A: Pred. Err. (%) | A: RE (%) | B: Pred. Err. | B: RE |
|---|---|---|---|---|---|---|
| ResNet-20 | 0.27 | 30.31 | x | x | x | x |
| ResNet-34 | 0.46 | 27.95 | x | x | x | x |
| ResNet-44 | 0.66 | 27.02 | 26.44 | 2.14 | 26.44 | 2.14 |
| ResNet-101 | 1.7 | 23.86 | 23.39 | 1.96 | 22.91 | 3.98 |

Table 2: Prediction of the top-1 test error for ResNet models on the CIFAR-100 dataset, whose validation set contains approximately 5.85% label errors Northcutt et al. (2021). The number of parameters and corresponding top-1 test errors were obtained from the original paper Jafar and Lee (2023). Note that the number of parameters in this table differs from those in Table 1 due to variations in the width of each layer and other adjustments. The relative errors for the ResNet-44 and ResNet-101 models are both less than 5%.

## III. VGGs on ImageNet:

In our analysis of the VGG family of models tested on ImageNet, we first examine the least-squares regression lines presented on a log-log scale. The regression lines adhere to scaling laws, displaying slopes of -1.21 for models that utilize Batch Normalization (BN) and -1.11 for those that do not. This difference in slope values indicates that the BN model is slightly more efficient with its parameters than the one without BN, suggesting a faster rate of decrease in test error.

The VGG implementations on ImageNet exhibit a non-negligible training error. However, as long as the training error is comparably small to the test error, we can still apply Eq. (17) to obtain an estimate of the slope for smaller models. The values of $\alpha$ were found to be 0.551 for models without BN and 0.478 for those with BN. The slope for models without BN is -1.581, while the slope for models utilizing BN is -1.784. The values are slightly lower but qualitatively consistent with those obtained using the least-squares method for three observations. Tables 3 and 4 present the prediction performance results, and the performance differences shown in the tables primarily result from the implementation of BN. Since there are only three observations, both methods yield identical predictions.

| Model-Depth | Size (M) | Test Error (%) | Pred. Test Error (%) | Relative Error (%) |
|---|---|---|---|---|
| VGG-13 | 133 | 30.1 | x | x |
| VGG-16 | 138 | 28.4 | x | x |
| VGG-19 | 144 | 27.6 | 26.55 | 3.8 |

Table 3: Prediction of the top-1 test error for VGG models without Batch Normalization on the ImageNet dataset. The number of parameters and top-1 test errors were obtained from the original paper by Simonyan and Zisserman (2014) as well as various benchmark re-implementations, including those in PyTorch and TensorFlow Paszke et al. (2019); PyTorch Development Team (2025). There may be slight variations depending on the training setup, such as data augmentation techniques and learning rate schedules.

| Model-Depth | Size (M) | Test Error (%) | Pred. Test Error (%) | Relative Error (%) |
|---|---|---|---|---|
| VGG-13-BN | 133 | 28.4 | x | x |
| VGG-16-BN | 138 | 26.6 | x | x |
| VGG-19-BN | 144 | 25.8 | 24.66 | 4.4 |

Table 4: Prediction of the top-1 test error for the VGG models with Batch Normalization on the ImageNet dataset Paszke et al. (2019); PyTorch Development Team (2025). The original VGG paper did not incorporate BN, which was introduced later by Ioffe and Szegedy (2015). There may be slight variations depending on the training setup, such as data augmentation techniques and learning rate schedules.

### IV. InceptionNets on Udacity:

The paper by Nadella et al. (2024) measures the performance of Inception architectures on the Kaggle SAP (Steering Angle Prediction) dataset of approximately 97,330 road images and the corresponding human steering angles. The images were captured from a dashcam mounted on a real car, recorded during actual driving sessions at 30 fps under varying real-world driving conditions including different lighting conditions and road types. The task is a regression problem that predicts the continuous-valued steering angle from road images. While our analysis is primarily designed for classification, we apply the proposed method to the steering angle prediction task to demonstrate its robustness and generalizability. The MSE is computed by averaging the squared difference between the predicted and actual steering angle over all test images, and serves as the mean prediction error reported in the following table.

| Model | Size (M) | MSE | A: Pred. MSE | A: RE (%) | B: Pred. MSE | B: RE |
|---|---|---|---|---|---|---|
| InceptionNet | 13 | 0.06044 | x | x | x | x |
| InceptionNet a | 14.5 | 0.05945 | x | x | x | x |
| InceptionNet b | 15.5 | 0.05849 | 0.05885 | 0.62 | 0.05885 | 0.62 |
| InceptionNet c | 17 | 0.05846 | 0.057649 | 1.39 | 0.05805 | 0.7 |
| InceptionNet d | 17.6 | 0.05961 | 0.05815 | 2.5 | 0.05775 | 3.12 |
| InceptionNet e | 20.7 | 0.05802 | 0.05817 | 0.26 | 0.05636 | 2.86 |
| InceptionNet f | 21.7 | 0.05907 | 0.05761 | 2.47 | 0.05602 | 5.16 |

Table 5: Predicted MSE for the InceptionNet family of models on the Udacity dataset. The box-module dimension $\log_2(1 + \alpha)$, estimated from the first two models, is 6.612.

## 5 Analysis of primary assumptions with random balls

Our earlier analysis in Lemma 2 relies on two primary assumptions. We carefully assess the conditions needed to validate these assumptions and explore their connections to the concentration results that underpin the generalization error bounds. In particular, we address the probability that random balls of a fixed radius cover the unit $d_e$-ball densely, where $d_e$ is the effective dimension adopted here to address the curse of dimensionality in data sampling. Furthermore, we derive the concentration bound necessary for the histogram approximation assumption.

We use the random process $X(\beta_l)$ to generate centers for balls of radius $\beta_l$, one at a time. These centers are drawn independently based on the density function $\mathcal{P}$. We denote $S$ as the set of centers generated by this process, and $X_{min}(S)$ as the smallest subset of $S$ that forms a covering of the input domain with balls of radius $\beta_l$.

The first lemma below discusses the probability of a sequence covering the unit $d_e$-ball randomly for the first time, while the following lemma focuses on a concentration inequality to approximate the unknown density function using histogram approximation. The proofs of the first lemma and the second lemma are provided in Appendices D and E, respectively.

**Lemma 3.** *Assume the density function $\mathcal{P} \geq k > 0$ for some $k$ over $B_{d_e}(\mathbf{0}, 1)$. The probability of the initial $N \geq 1/(c\beta_l^{d_e})$ samples in $X(\beta_l)$ forming a cover of the unit $d_e$-ball is*

$$P(|X_{\min}| \leq N) \geq 1 - \frac{1}{cN\beta_l^{d_e}}, \tag{18}$$

*for some constant $c$ satisfying $P(\mathbf{x}, B_{d_e}(\mathbf{x}, \beta_l)) \geq c\beta_l^{d_e}$ for any $\mathbf{x} \in B_{d_e}(\mathbf{0}, 1)$ (see Eq. (7)).* ∎

The classification network $\mathcal{M}$ divides the input domain into non-overlapping regions and assigns each region a specific class. The collection of all these regions is referred to as $R_s$. We next address the concentration inequality to approximate the following term with the histogram for region $p \in R_s$:

$$\int_p P(\mathbf{x}) \, d\mathbf{x}. \tag{19}$$

In histogram approximation, $\int_p P(\mathbf{x})d\mathbf{x}$ is replaced with $\frac{n_p}{N}$, where $n_p$ is the number of training points assigned to region $p$ among $N$ total, and is a random variable determined by the sampling sequence.

**Lemma 4.** *Recall the process $X(\beta_l)$. Let $S$ be a sequence of $N$ points that forms a $\beta_l$-covering of the input domain, with $|X_{min}(S)| \leq N$. Then the following concentration inequality holds for the histogram approximation of $\int_p \mathcal{P}(\mathbf{x})d\mathbf{x}$ over any bin $p$:*

$$P(|\int_p \mathcal{P}(\mathbf{x})d\mathbf{x} - \frac{n_p(S)}{N}| \geq t_p) \leq 2\exp(-2Nt_p^2). \tag{20}$$

*If $t_p$ goes to zero at a rate slower than $N$ goes to infinity, then in the limit as $N \to \infty$, we have $\int_p \mathcal{P}(\mathbf{x})\,d\mathbf{x} = \frac{n_p(S)}{N}$.* ∎

**Proposition 2.** *Assume $\mathcal{P} \geq k > 0$ for some $k$ over $B_d(\mathbf{0}, 1)$. Consider the random ball model $X(\beta_l)$, the assumptions of $\beta_l$-covering and the histogram approximation hold for any $\beta_l > 0$, when $N \to \infty$.* ∎

**Proof.** Using $P(A, B) = P(A)P(B|A)$ where $A = |X_{min}| \leq N$ denotes the $\beta_l$-covering of the input domain and $B = |\int_p \mathcal{P}(\mathbf{x})d\mathbf{x} - \frac{n_p}{N}| < t_p$ for any $t_p > 0$.

According to Lemma 3, $P(|X_{min}| \leq N) \geq 1$, as $N \to \infty$. Hence, the assumption of $\beta_l$-covering is affirmative. In addition, Lemma 4 asserts that the Hoeffding's inequality as described in Eq. (20) indicates that for any region and any $t_p > 0$ and $N \to \infty$,

$$P(|\int_p \mathcal{I}(\mathbf{x} \in p)\mathcal{P}(\mathbf{x})d\mathbf{x} - \frac{n_p}{N}| < t_p \,\big|\, |X_{min}| \leq N) > 1. \tag{21}$$

This means that $\int_p \mathcal{P}(\mathbf{x})d\mathbf{x} = \frac{n_p}{N}$. Hence $P(A, B) = 1$ as $N \to \infty$.  Q.E.D.

**Remark.** In practice, it is useful to estimate the sample size required for the two primary assumptions to hold with high probability. We thus provide an $(\epsilon, \delta)$ PAC-type analysis to characterize this sample size requirement.

Suppose $N$ training points form a $\beta_l$-covering of $B_{d_e}(\mathbf{0}, 1)$. We let $R_s$ denote the decision regions of a network $\mathcal{M}$. Then, by a volume argument using balls of radius $\beta_l$ to cover the input domain, we obtain $N\beta_l^{d_e} \geq 1$.

Combining this equation with Eq. (12), where $\frac{1}{\#W^{1/\log_2(1+\alpha(\gamma_s))}} \geq \gamma_s$, and letting $\nu = \beta_l/\gamma_s \leq 1$ denote the scaling factor of the decision partition of the network $\mathcal{M}$, we obtain

$$\frac{1}{\#W^{1/\log_2(1+\alpha(\gamma_s))}} \geq \gamma_s = \frac{\beta_l}{\nu} \geq \frac{1}{\nu}\left(\frac{1}{N}\right)^{1/d_e}. \tag{22}$$

Hence,

$$N \geq \frac{1}{\nu^{d_e}}\#W^{d_e/\log_2(1+\alpha)} \geq |R_s|. \tag{23}$$

The last inequality is a necessary condition for at least one training point in each decision region.

For a single decision region $p \in R_s$, the probability $\int_p \mathcal{P}(\mathbf{x})d\mathbf{x} \approx \frac{n_p}{N}$ can be estimated using Hoeffding's inequality,

$$P(|\int_p \mathcal{P}(\mathbf{x})d\mathbf{x} - \frac{n_p}{N}| > \epsilon) \leq 2e^{-2N\epsilon^2}. \tag{24}$$

The probability estimate is only "accurate" if all decision regions are approximated accurately simultaneously. Applying the union bound and substituting the single-bin bound, with failure probability at most $\delta$, we obtain

$$\delta = P\left(\exists q \in R_s : |\int_q \mathcal{P}(\mathbf{x})d\mathbf{x} - \frac{n_q}{N}| > \epsilon\right) \leq \sum_{p \in R_s} P(|\int_p \mathcal{P}(\mathbf{x})d\mathbf{x} - \frac{n_p}{N}| > \epsilon) \leq 2|R_s|e^{-2N\epsilon^2}. \tag{25}$$

Now, we apply Eq. (23) to bound $|R_s|$ in the above failure probability:

$$\delta \leq \frac{2}{\nu^{d_e}}\#W^{d_e/\log_2(1+\alpha)}e^{-2N\epsilon^2}. \tag{26}$$

Consequently, under the assumptions that the training points form a $\beta_l$-covering of $B_{d_e}(\mathbf{0},1)$ and each decision region contains at least one training point, we obtain

$$N \leq \frac{1}{2\epsilon^2}\left(\ln\left(\#W^{d_e/\log_2(1+\alpha)}\right) + \ln\frac{2}{\delta} + \ln\frac{1}{\nu^{d_e}}\right). \tag{27}$$

## 6 Conclusions

Our analysis of scaling laws for classification networks focuses on three key considerations: the covering radius of training-point balls, the histogram approximation of the regional average of the unknown density function, and the relationship between ball radius and network parameters. This approach differs from studies focused on the interpolation zone, which has been the predominant setting for scaling law research. We present the conditions for the emergence of scaling laws and demonstrate that the scaling laws for sample size and network size depend on different factors. The former depends on the effective dimension of the data manifold, independent of the network model and learning method. For the latter, we introduce the box-modulus dimension, which measures the rate at which network parameters accumulate as one zooms into the classification boundary. This dimension yields a scaling law that is dependent on both the model and the learning method, and we use it to estimate the resources needed to increase model depth and adjust parameter settings to achieve a target generalization performance.

In light of these findings, we propose the following question: Can we extend our analysis, within a PAC framework, to jointly characterize network behavior in terms of generalization error, number of parameters, and the number of training points? By doing so, we may uncover additional patterns or principles that govern network performance, such as the conditions under which the scaling laws break down or no longer hold, thereby deepening our understanding of the complex behaviors observed empirically.

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

## A   Proof of Lemma 2

One-hot classifiers are piecewise constant functions, assigning inputs to a single class $i$ within the range of $\{1, \cdots, l\}$. The standard coordinate basis $\mathbf{e}_i$ represents class $i$ in a one-hot coding system. To compute the $0/1$ error function, we can use the formula below: for any $f(\mathbf{y}), f(\mathbf{z}) \in \{\mathbf{e}_1, \cdots, \mathbf{e}_l\}$,

$$g(f(\mathbf{y}), f(\mathbf{z})) = \frac{1}{\sqrt{2}} \|f(\mathbf{y}) - f(\mathbf{z})\|_2 = \begin{cases} 1 \text{ if } f(\mathbf{y}) \neq f(\mathbf{z}) \\ 0 \text{ otherwise.} \end{cases} \tag{28}$$

For a given classifier $f$, a ball exists around any point $\mathbf{x}$ in the input domain, where all the points inside this ball have the same class as that of $\mathbf{x}$. If $\mathbf{x}$ lies on the classification boundary of $f$, the radius of the ball is zero. We use the notation $\eta_f(\mathbf{x}) \geq 0$ to denote the maximum radius of the ball centered at $\mathbf{x}$ such that $f(\mathbf{y}) = f(\mathbf{x})$ for all $\mathbf{y}$ belonging to $B_d(\mathbf{x}, \eta_f(\mathbf{x}))$. Therefore, the class of $f$ at any point $\mathbf{y}$ is the same as that of $\mathbf{x}$ when the distance between $\mathbf{x}$ and $\mathbf{y}$ is less than or equal to $\eta_f(\mathbf{x})$. Hence, we have

$$h_f(\mathbf{x}; \|\mathbf{x} - \mathbf{y}\|_2) = g(f(\mathbf{x}), f(\mathbf{y})) = 0 \text{ if } \|\mathbf{x} - \mathbf{y}\|_2 \leq \eta_f(\mathbf{x}). \tag{29}$$

We can bound $h_f$ with the following radius function:

$$\hat{h}_f(\mathbf{x}; \|\mathbf{x} - \mathbf{y}\|_2) = \begin{cases} 1 \text{ if } \|\mathbf{x} - \mathbf{y}\|_2 > \eta_f(\mathbf{x}) \\ 0 \text{ otherwise.} \end{cases} \geq h_f(\mathbf{x}; \|\mathbf{x} - \mathbf{y}\|_2). \tag{30}$$

$\hat{h}_f(\mathbf{x}; \|\mathbf{x} - \mathbf{y}\|_2)$ is a non-decreasing function of $\|\mathbf{x} - \mathbf{y}\|_2$ for any function $f$. In the case where $\mathbf{x}$ is on the classification boundary, $\hat{h}_f(\mathbf{x}; \|\mathbf{x} - \mathbf{y}\|_2)$ will always be equal to 1 for any value of $\mathbf{y} \neq \mathbf{x}$.

The classification network $\mathcal{M}$ is responsible for learning the classifier function $f$ by utilizing $N$ training data pairs $(\mathbf{x}_i, f(\mathbf{x}_i))$. The function of $\mathcal{M}$ is piece-wise constant, dividing the input domain into non-overlapping regions and assigning each region a specific class. The collection of all these regions is referred to as $R_s$. Each region in $R_s$ is associated with two geometric parameters related to balls: the smallest enclosing radius and the largest inscribed radius. The smallest enclosing radius that encompasses all regions in $R_s$ and the largest inscribed radius for all regions in $R_s$ are denoted as $\gamma_s$ and $\beta_l$, respectively.

The training points are $\beta_l$-covering distributed means that every region of $R_s$ must have at least one training point (see Lemma 1). For any input point $\mathbf{x}$ located in region $p$, let $\mathbf{x}_p$ represent a training point within that region. The distance between $\mathbf{x}$ and $\mathbf{x}_p$ will always be less than or equal to $2\gamma_s$, since the radius of the enclosing ball for the region is $\gamma_s$. To calculate the error between $f(\mathbf{x})$ and $\mathcal{M}(\mathbf{x})$ with respect to $\mathbf{x}_p$, we can use the following equation:

$$\begin{aligned} g(f(\mathbf{x}), \mathcal{M}(\mathbf{x})) &= \frac{1}{\sqrt{2}}(\|f(\mathbf{x}) - \mathcal{M}(\mathbf{x}) + f(\mathbf{x}_p) - f(\mathbf{x}_p) + \mathcal{M}(\mathbf{x}_p) - \mathcal{M}(\mathbf{x}_p)\|_2) \\ &\leq \frac{1}{\sqrt{2}}(\|f(\mathbf{x}) - f(\mathbf{x}_p)\|_2 + \|f(\mathbf{x}_p) - \mathcal{M}(\mathbf{x}_p)\|_2 + \|\mathcal{M}(\mathbf{x}_p) - \mathcal{M}(\mathbf{x})\|_2) \\ &\leq g(f(\mathbf{x}), f(\mathbf{x}_p)) + \varepsilon_p. \end{aligned} \tag{31}$$

Here, we define $\frac{1}{\sqrt{2}}\|f(\mathbf{x}_p) - \mathcal{M}(\mathbf{x}_p)\|_2$ as $g(f(\mathbf{x}_p), \mathcal{M}(\mathbf{x}_p)) = \varepsilon_p$. As points in the same region of $R_s$ of $\mathcal{M}$ have the same class, the term $\|\mathcal{M}(\mathbf{x}_p) - \mathcal{M}(\mathbf{x})\|_2$ is zero.

Denote $\{\mathbf{x}_{p,i}\}_{i=1}^{n_p}$ as the set of $n_p$ training points in region $p$. By applying Eq. (31) to each training point in the region and then computing the average, we can obtain the following result:

$$\begin{aligned} g(f(\mathbf{x}), \mathcal{M}(\mathbf{x})) &\leq \frac{1}{n_p} \sum_{i=1}^{n_p} g(f(\mathbf{x}), f(\mathbf{x}_{p,i})) + \bar{\varepsilon}_p \\ &\leq \max_i g(f(\mathbf{x}), f(\mathbf{x}_{p,i})) + \bar{\varepsilon}_p. \end{aligned} \tag{32}$$

To assess the expected error for probability distribution $\mathcal{P}$, we use Eq. (32) for each point $\mathbf{x}$ in a region $p$, and then sum over all regions of $R_s$ to obtain:

$$\begin{aligned} \int_{B_d(\mathbf{0},1)} g(f(\mathbf{x}), \mathcal{M}(\mathbf{x}))\mathcal{P}(\mathbf{x}) \, d\mathbf{x} &= \sum_{p \in R_s} \int_p g(f(\mathbf{x}), \mathcal{M}(\mathbf{x}))\mathcal{P}(\mathbf{x}) \, d\mathbf{x} \\ &\leq \sum_{p \in R_s} \int_p \max_i g(f(\mathbf{x}), f(\mathbf{x}_{p,i}))\mathcal{P}(\mathbf{x})d\mathbf{x} \\ &+ \sum_{p \in R_s} \bar{\varepsilon}_p \int_p \mathcal{P}(\mathbf{x}) \, d\mathbf{x}. \end{aligned} \tag{33}$$

In the equation, $\bar{\varepsilon}_p$ represents the mean of the prediction error, which is a constant in region $p$, and the equality holds because $R_s$ partitions the input domain. The left side indicates the generalization error, while the right comprises two integrals. The first integral is the maximum value of the function g calculated with respect to all training points $\mathbf{x}_{p,i}$ in region $p$. The second term in Eq. (33) can be further derived using the

assumption of histogram approximation to relate this term to the average of all training points:

$$\sum_{p \in P_s} \bar{\varepsilon}_p \int_p \mathcal{P}(\mathbf{x}) \, d\mathbf{x} = \sum_{p \in P_s} \bar{\varepsilon}_p \frac{n_p}{N} = \frac{1}{N} \sum_{p \in P_s} \sum_{i=1}^{n_p} \|f(\mathbf{x}_{p,i}) - \mathcal{M}(\mathbf{x}_{p,i})\|_2$$

$$= \frac{1}{N} \sum_{i=1}^{N} \|f(\mathbf{x}_i) - \mathcal{M}(\mathbf{x}_i)\|_2. \tag{34}$$

Thus, Eq. (33) can be expressed as:

$$\int_{B_d(\mathbf{0},1)} g(f(\mathbf{x}), \mathcal{M}(\mathbf{x})) \mathcal{P}(\mathbf{x}) \, d\mathbf{x} \leq \sum_{p \in R_s} \int_p \max_i g(f(\mathbf{x}), f(\mathbf{x}_{p,i})) \mathcal{P}(\mathbf{x}) d\mathbf{x}$$

$$+ \frac{1}{N\sqrt{2}} \sum_i \|f(\mathbf{x}_i) - \mathcal{M}(\mathbf{x}_i)\|_2$$

$$= \sum_{p \in R_s} \int_p \max_i g(f(\mathbf{x}), f(\mathbf{x}_{p,i})) \mathcal{P}(\mathbf{x}) d\mathbf{x}$$

$$+ \frac{1}{N\sqrt{2}} \sum_i g(f(\mathbf{x}_i), \mathcal{M}(\mathbf{x}_i)). \tag{35}$$

We can utilize two facts to calculate an upper bound for the first term in Eq. (35). The first is that $\|\mathbf{x} - \mathbf{x}_{p,i}\|_2 \leq 2\gamma_s$, and the second is that $\hat{h}_f$ is a non-decreasing function, as per Eqs. (30) and (29). By using these facts, we obtain the following inequality:

$$\sum_{p \in R_s} \int_p \max_i g(f(\mathbf{x}), f(\mathbf{x}_{p,i})) \mathcal{P}(\mathbf{x}) d\mathbf{x} \leq \sum_{p \in R_s} \int_p \max_i \hat{h}_f(\mathbf{x}; \|\mathbf{x} - \mathbf{x}_{p,i}\|_2) \mathcal{P}(\mathbf{x}) d\mathbf{x}$$

$$\leq \int \hat{h}_f(\mathbf{x}; 2\gamma_s) \mathcal{P}(\mathbf{x}) d\mathbf{x}$$

$$\leq \int \mathbf{1}([2\gamma_s \geq \eta_f(\mathbf{x})]) \mathcal{P}(\mathbf{x}) d\mathbf{x}. \tag{36}$$

Here, $\mathbf{1}([2\gamma_s \geq \eta_f(\mathbf{x})])$ is an indicator function for $2\gamma_s \geq \eta_f(\mathbf{x})$, and the last inequality is a result of Eq. (30). For function $f$, the value of $\eta_f(\mathbf{x})$ remains fixed. Moreover, the function $\mathbf{1}([2\gamma_s \geq \eta_f(\mathbf{x})])$ is an increasing function of $\gamma_s$. Therefore, a network $\mathcal{M}$ with a lower value of $\gamma_s$ can lead to a reduced generalization bound.

(i) The probability mass that contributes to $\int \mathbf{1}([2\gamma_s \geq \eta_f(\mathbf{x})]) \mathcal{P}(\mathbf{x}) d\mathbf{x}$, which can generate generalization error, can be bounded. The points satisfying $2\gamma_s \geq \eta_f(\mathbf{x})$ are within a distance no greater than $2\gamma_s$ from the classification boundary of $f$, denoted by $\partial f$, as the schematic diagram illustrated in Figure 1(b). The boundary can be decomposed as the union of semi-algebraic subsets as follows:

$$\partial f = \bigcup_{k=0}^{d-1} f_k = \bigcup_{k=0}^{d-1} \bigcup_i f_k^i,$$

where $f_k^i$ is a $k$-dimensional semi-algebraic set. We define the $2\gamma_s$-neighborhood of $\partial f$ in $\ell_2$-norm as

$$A(2\gamma_s, \partial f) = \{\mathbf{x} \in B_d(\mathbf{0}, 1) \mid \mathrm{dist}(\mathbf{x}, \partial f) \leq 2\gamma_s\} = \bigcup_{k=0}^{d-1} A(2\gamma_s, f_k) \tag{37}$$

and the $2\gamma_s$-neighborhood of $f_k = \bigcup_i f_k^i$ is defined as

$$A(2\gamma_s, f_k) = \bigcup_i A(2\gamma_s, f_k^i) = \bigcup_i \{\mathbf{x} \in B_d(\mathbf{0}, 1) \mid \mathrm{dist}(\mathbf{x}, f_k^i) \leq 2\gamma_s\}.$$

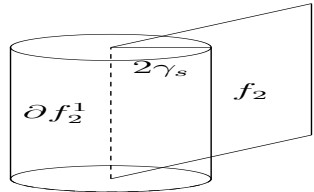

Figure 3: The boundary of closed surface $f_2$ consists of four sub-components under stratification, and its dimension is 2. The central axis (represented as a dashed line segment) of the cylinder in $\mathbb{R}^3$ is the first part of the boundary of $f_2$, denoted as $\partial f_2^1$. This axis has a dimension of 1, while the radius of the cylinder is $2\gamma_s$. The volume of the cylinder can be expressed as $|\partial f_2^1|(2\gamma_s)^2\pi$.

The volume of $A(2\gamma_s, f_k)$ can be bounded in accordance with the union bound and $\dim f_k^j = k$ as

$$|A(2\gamma_s, f_k)| \leq \sum_i |A(2\gamma_s, f_k^i)| \leq (2\gamma_s)^{d-k}|B_{d-k}(\mathbf{0}, 1)| \sum_i |f_k^i|$$
$$= (2\gamma_s)^{d-k}|B_{d-k}(\mathbf{0}, 1)||f_k|. \tag{38}$$

See the illustrative example on calculating $|A(2\gamma_s, f_2)|$ in Figure 3. In the above, we utilize the fact that $|A(2\gamma_s, f_k^i)|$ equals $|B_{d-k}(\mathbf{0}, 2\gamma_s)||f_k^i|$, where $|B_{d-k}(\mathbf{0}, 2\gamma_s)| = (2\gamma_s)^{d-k}|B_{d-k}(\mathbf{0}, 1)|$.

By applying the union bound to Eq. (37) and then to Eq. (38), we derive a bound for the volume of the $2\gamma_s$-neighborhood of $\partial f$:

$$|A(2\gamma_s, \partial f)| \leq \sum_{k=0}^{d-1} |A(2\gamma_s, f_k)| \leq \sum_{k=0}^{d-1} (2\gamma_s)^{d-k}|B_{d-k}(\mathbf{0}, 1)||f_k|.$$

Consequently,

$$\int 1([2\gamma_s \geq \eta_f(\mathbf{x})])\mathcal{P}(\mathbf{x})d\mathbf{x} = c_{\mathcal{P}}|A(2\gamma_s, \partial f)| \leq c_{\mathcal{P}} \sum_{k=0}^{d-1} (2\gamma_s)^{d-k}|B_{d-k}(\mathbf{0}, 1)||f_k|. \tag{39}$$

The first equality bounds the probability mass by the volume of the points over the $2\gamma_s$-neighborhood of $\partial f$, where $c_p \in (0, 1)$ represents the average density of $\mathcal{P}$ distributed across the neighborhood.

(ii) When $\gamma_s$ is small and $|f_{d-1}| \neq 0$, the sum $\sum_{k=0}^{d-1}(2\gamma_s)^{d-k}|B_{d-k}(\mathbf{0}, 1)||f_k|$ in equation (39) can be approximated by the leading term $(2\gamma_s)|B_1(\mathbf{0}, 1)||f_{d-1}| = 4\gamma_s|f_{d-1}|$ with $|B_1(\mathbf{0}, 1)| = 2$ (the length of a segment).

## B    Box-module dimensions of shallow and deep networks

We develop DNNs by increasing depth, in contrast to the wider configurations discussed by Sharma and Kaplan (2022).

Assume that the training points are distributed within a unit cube $[0, 1]^{d_e}$, where $d_e$ denotes the effective dimension. The unit cube can be structured as a grid composed of hypercubes, each with a side length of $s < 1$. The number of hypercubes, denoted as $N_s$, is given by $(s)^{-d_e}$. Sharma and Kaplan (2022) suggest that the number of parameters $\#S_s$ of a network $\mathcal{S}_s$ required to realize this grid is proportional to $N_s$.

Analogously, if the side length of hypercubes in the grid becomes $2s < 1$, the number of hypercubes is $N_{2s} = (2s)^{-d_e}$, and the number of parameters $\#S_{2s}$ of a network to realize this grid is proportional to $N_{2s}$.

Following this reasoning, when reducing the side length from $2s$ to $s$, the number of parameters required for the network $\mathcal{S}_s$ is given by $\#S_s = c(s)^{-d_e} = 2^{d_e}\#S_{2s}$, where $c$ is the proportionality constant. Thus, halving the side length increases the required number of parameters by a factor of $2^{d_e}$.

This exponential increase in parameter counts results from widening the network while keeping the depth fixed. However, we argue that the parameters used to realize the grid with side length $2s$ can be reused

to construct the finer grid with side length $s$, which the approach of Sharma and Kaplan (2022) does not consider.

Alternatively, the grid configuration with side length $s$ can be constructed from two grids with side length $2s$, where the second is obtained by translating the first by $s$ units along each axis. If a network realizing a grid of side length $2s$ can be reused and augmented with this translation, the number of parameters required for the finer grid with side length $s$ can be substantially reduced compared to training a new network from scratch.

Suppose we have a sub-network $\mathcal{N}_{2s}^1$ that consists of $\frac{1}{2s}$ parallel and equally spaced hyperplanes with respect to the first component of the coordinate system of $d_e$ components. From those hyperplanes, we can generate $\frac{1}{s}$ parallel and equally spaced hyperplanes by adding a layer to $\mathcal{N}_{2s}^1$ as follows:

$$\mathcal{N}_s^1 = \text{ReLU}\left[\begin{array}{c}\mathcal{N}_{2s}^1 \\ \mathcal{N}_{2s}^1 + \begin{bmatrix}-s\\0\\\vdots\\0\end{bmatrix}\end{array}\right] = \text{ReLU } \mathbf{A}_s \mathcal{N}_{2s}^1. \tag{40}$$

The upper component in $\mathbf{A}_s$ is the identity matrix $\mathbf{I}$, which generates the same hyperplanes as those in $\mathcal{N}_{2s}^1$. The bottom mapping involves translating by $-s$ along the first coordinate axis, resulting in another set of hyperplanes, each of which is shifted by $s$ units toward the origin relative to the corresponding hyperplane in $\mathcal{N}_{2s}^1$. The ReLU layer clips the output to $[0,1]^{d_e}$.

By concatenating the outputs of $\mathcal{N}_{2s}^i$ for $i = 1, \cdots, d_e$, we obtain the network that generates the grid configuration of side length $s$:

$$\mathcal{N}_s = \begin{bmatrix}\mathcal{N}_{2s}^1 \\ \vdots \\ \mathcal{N}_{2s}^{d_e}\end{bmatrix} = \text{ReLU}\left(\text{diag}(\mathbf{A}_s, \cdots, \mathbf{A}_s)\right)\mathcal{N}_{2s}, \tag{41}$$

where $\text{diag}(\mathbf{A}_s, \cdots, \mathbf{A}_s)$ is the block diagonal matrix with each block $\mathbf{A}_s$. $\mathcal{N}_s$ produces a grid partition of side length $s$ over the input domain $[0,1]^{d_e}$. Figure 4 shows how the network $\mathcal{N}_s$ is constructed within the domain $[0,1]^2$, as described in Eqs. (40) and (41).

This construction shows that adding a layer increases the number of parameters linearly with respect to $d_e$. This is captured by $\#W_s = \#W_{2s} + d_e c \leq (1+\alpha)\#W_{2s}$, where $c$ denotes the number of parameters for $\mathbf{A}_s$, $\alpha > 0$, and $\#W_s$ and $\#W_{2s}$ denote the number of parameters of $\mathcal{N}_s$ and $\mathcal{N}_{2s}$, respectively. This linear increase in parameter counts stands in sharp contrast to the exponential growth observed in the network of Sharma and Kaplan (2022).

Next, we compare the mean absolute test errors. Assume that each partition region of $\mathcal{N}_s$ in $[0,1]^{d_e}$ contains at least one training point. Let $\mathbf{x}$ and $\mathbf{x}_i$ be two points within a hypercube of side length $s$, with $\|\mathbf{x} - \mathbf{x}_i\| \leq \sqrt{d_e}$. By the geometry of the hypercube, $\sqrt{d_e}s$ represents the maximum distance between two points inside a hypercube of side length $s$.

Define $f : \mathbb{R}^{d_e} \to \mathbb{R}$ as the target function with Lipschitz constant $K_f$, where $|f(\mathbf{x}) - f(\mathbf{y})| \leq K_f\|\mathbf{x} - \mathbf{y}\|_2$. We suppose $\mathcal{N}_s$ interpolates $f$ at the training points; i.e., $\mathcal{N}_s(\mathbf{x}_i) = f(\mathbf{x}_i)$. Since $\mathcal{N}_s$ is a piecewise constant function over a partition of hypercubes and each hypercube contains at least one training point, if $\mathbf{x}$ and $\mathbf{x}_i$ are in the same hypercube, then $\mathcal{N}_s(\mathbf{x}) = \mathcal{N}_s(\mathbf{x}_i)$. This leads us to the bound:

$$|\mathcal{N}_s(\mathbf{x}) - f(\mathbf{x})| = |\mathcal{N}_s(\mathbf{x}_i) - f(\mathbf{x})| = |f(\mathbf{x}_i) - f(\mathbf{x})| \leq K_f\|\mathbf{x}_i - \mathbf{x}\|_2 \leq K_f\sqrt{d_e}s.$$

Consequently, if we let $2^l s = 1$, where $l$ denotes the number of refinement levels, the test error can be bounded by:

$$\int_{[0,1]^{d_e}} |f(\mathbf{x}) - \mathcal{N}_s(\mathbf{x})|d\mathbf{x} \leq K_f\sqrt{d_e}s = K_f\sqrt{d_e}(1/2^l). \tag{42}$$

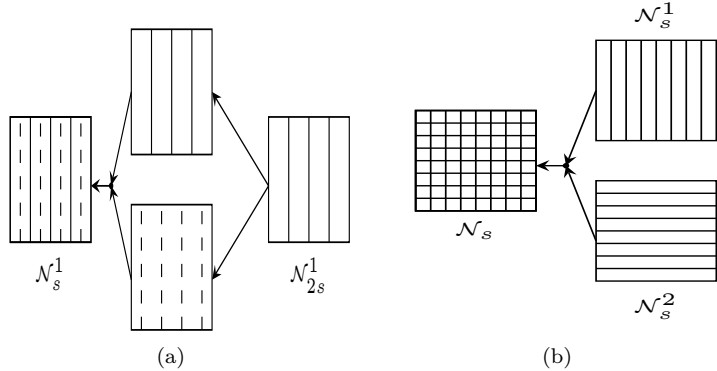

Figure 4: Schematic diagrams in (a) and (b) illustrate how the network $\mathcal{N}_s$ is formed by adding an additional layer to $\mathcal{N}_{2s}$, with its domain in $[0,1]^2$. Subfigure (a) depicts the configurations along the first coordinate, transitioning from $\mathcal{N}_{2s}^1$ to $\mathcal{N}_s^1$. Here, the distance between the parallel hyperplanes in $\mathcal{N}_{2s}^1$ is $2s$, while the distance for the hyperplanes in $\mathcal{N}_s^1$ is $s$. The middle top subfigure corresponds to $\mathcal{N}_{2s}^1$, whereas the middle bottom subfigure shows the configuration of $\mathcal{N}_{2s}^1 + [-s\ 0]^\top$. The configuration of $\mathcal{N}_s^1$ is obtained by concatenating those in $\mathcal{N}_{2s}^1$ and $\mathcal{N}_{2s}^1 + [-s\ 0]^\top$, as defined by Eq. (40). Subfigure (b) illustrates how the configurations of $\mathcal{N}_s^1$ and $\mathcal{N}_s^2$ are concatenated to obtain the configuration of $\mathcal{N}_s$, as defined by Eq. (41). The domain partition of $\mathcal{N}_s$ corresponds to the Cartesian product of the domain partitions of $\mathcal{N}_s^1$ and $\mathcal{N}_s^2$, as stated by Hwang and Tung (2023).

By unfolding the recurrence equation $\#W_s \leq (1+\alpha)\#W_{2s}$ $l$ times and using the boundary condition $\#W_1 = 1$, corresponding to a single hypercube over $[0,1]^{d_e}$, we can determine the number of parameters for network $\mathcal{N}_s$ as follows:

$$\#W_s \leq (1+\alpha)^l \#W_1 \leq (1+\alpha)^l. \tag{43}$$

If we substitute $2^l = (1+\alpha)^{l/\log_2(1+\alpha)}$ into Eq. (43), we obtain: $2^l \geq \#W_s^{1/\log_2(1+\alpha)}$. Then, if we substitute this expression for $2^l$ into Eq. (42), the test error for the network $\mathcal{N}_s$ is bounded by:

$$\frac{K_f \sqrt{d_e}}{\#W_s^{1/\log_2(1+\alpha)}}. \tag{44}$$

The absolute value of the slope in the log-log plot is $1/\log_2(1+\alpha)$.

In contrast, using the recurrence for the network of Sharma and Kaplan (2022): $\#S_s = 2^{d_e}\#S_{2s}$, with boundary condition $\#S_1 = 1$, we obtain $\#S_s = 2^{ld_e}\#S_1 = 2^{ld_e}$. Substituting $2^l = (\#S_s)^{1/d_e}$ into Eq. (42), the test error for the network $\mathcal{S}_s$ is bounded by

$$\frac{K_f \sqrt{d_e}}{\#S_s^{1/d_e}}. \tag{45}$$

The slope's absolute value is $1/d_e$, as noted in Sharma and Kaplan (2022).

Additionally, we argue that $\#S_s \geq \#W_s$ because the wide network includes additional parameters without accounting for the values of the existing ones in previous depths. Consequently, based on Eqs. (44) and (45), we establish a comparison between box-module dimensions, resulting in the inequality:

$$\log_2(1+\alpha) \leq d_e.$$

**Remark.** A bound that requires neither interpolation at training points nor a $\gamma_s$-covering of the input domain, in contrast to Eq. (42), is given in Eq. (2.2) of Sharma and Kaplan (2022), where $L = \int_{[0,1]^{d_e}} (f(\mathbf{x}) - \mathcal{N}_s(\mathbf{x}))^2 d\mathbf{x} \lesssim K_f^2 s^2 d_e$. Intuitively, the error comes entirely from how much $f$ can vary within a single

hypercube. The Lipschitz condition bounds that variation by the diameter of the hypercube, $\sqrt{d_e}s$. The integral then averages this worst-case error over the whole domain, and since the domain has unit volume, the average equals the pointwise worst-case bound.

## C   Proof of Proposition 1

Because the two networks $[I + \rho M_L] \circ [I + \rho M_{L-1}] \circ \cdots \circ [I + \rho M_1]$ and $[\rho M_L] \circ [\rho M_{L-1}] \circ \cdots \circ [\rho M_1]$ share the same input domain partition Hwang and Tung (2023), we can simplify our notation for convenience. If our focus is on the input domain partition, we define:

$$\mathcal{M}_L^0 = [\rho M_L] \circ [\rho M_{L-1}] \circ \cdots \circ [\rho M_1].$$

We will construct $\mathcal{M}_l^0 = [\rho M_l] \cdots [\rho M_1]$ layer by layer, ensuring that it satisfies the condition in Eq. (12) with a constant value of $\alpha$ maintained throughout the network's depth.

The function of $\mathcal{M}_L^0$ is piece-wise continuous. Pascanu et al. (2013) previously discussed the machine's capability for function approximations, focusing on the number of regions generated for function approximation. If $P_l^0$ is the finest input domain partition of $\mathcal{M}_l^0$, where each partitioning region of $P_l^0$ contains no subregions, then $|P_l^0|$ represents the size of $P_l^0$. When considering $P_l^0$, each partitioning region is a polytope, an affine linear mapping domain. Additionally, we note that $P_{l+1}^0$ is a refined partition of $P_l^0$, meaning that every region in $P_{l+1}^0$ is within a region in $P_l^0$. This refinement of partitioning regions in $P_l^0$ sets up a tree-like partitioning of $P_l^0$, where each region in $P_{l+1}^0$ has a unique parent region in $P_l^0$.

Within $P_L^0$, every partitioning region forms a polytope, essentially the intersection of half-planes. The "diameter" of a region denotes the longest distance between two points within the region. The maximum diameter among all polytopes in a partition is equivalent to the diameter of the smallest enclosing ball of the partition. Calculating the diameter of a polytope according to Frieze and Teng (1994) can be intricate for polytopes of arbitrary dimension and facet number. Here, we are developing a tractable algorithm to partition a polytope and reduce the diameters of the resulting sub-polytopes to a constant fraction of the original value.

The algorithm begins by dividing a polytope $p$ in $P_L^0$ via a cutting hyperplane passing through an anchor point in $p$ into two sub-polytopes. This method is then applied recursively to each sub-polytope. As a result, polytope $p$ is the root of a tree-like structure, with intermediate and leaf nodes representing the sub-polytopes. The sub-polytopes at the leaf nodes that refine $p$ will have a diameter smaller than half that of the root polytope. Consequently, the collection of sub-polytopes at the leaf nodes of all trees forms the partition $P_{L+1}^0$, with the diameter to be less than half that of $P_L^0$.

The anchor point is the center of the maximum inscribed ellipsoid within the polytope. By Boyd and Vandenberghe (2004), we can determine this point using convex optimization methods. Additionally, we can utilize properties that approximate the volume of a polytope from both inside and outside at the anchor point, as described in Grötschel et al. (2012) to bound the radius of the smallest enclosing ball of a polytope in $P_{L+1}^0$. Let $\mathbf{a}^p \in \mathbb{R}^d$ be the anchor point of the polytope $p$. Any hyperplane passing through $\mathbf{a}^p$ divides the polytope into two sub-polytopes, with $q$ being one of them. The volume-reduction property shows a reduction in volume between $p$ and $q$, as demonstrated by the inequality

$$\text{vol}(q) \leq \left(1 - \frac{1}{d}\right) \text{vol}(p). \tag{46}$$

Additionally, if polytope $p$ is convex (or its convex hull if it's not), then the volume of $p$ can be effectively approximated from the outside by an ellipsoid enlarging a constant factor of the inner maximum inscribed ball of $p$; specifically, the outer ellipsoid of $p$ represents the boundary of the inner ellipsoid, expanded by a factor $d$ around the anchor point.

Let $\eta > 0$ represent the radius of the principal axis of the inner maximum volume inscribed ellipsoid of $p$, and $\tilde{\eta} > 0$ denote the smallest radius of this ellipsoid. Assume the relationship $\tilde{\eta}/\eta \geq \zeta > 0$ holds, where $\zeta$ is a predetermined value. Then, we can express the approximation of polytope volume from inside and outside by balls in the following inequality:

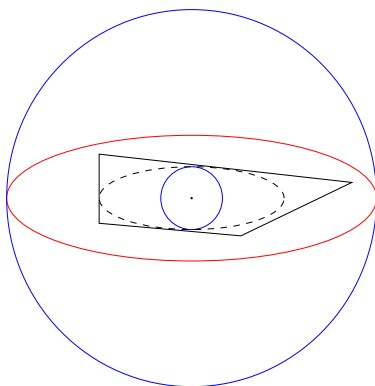

Figure 5: The dashed ellipsoid represents the maximum inscribed ellipsoid of polygon $p$. As per Eq. (47), the primary axis of the ellipsoid has a radius of $\eta$, while the minor axis has a radius of $\tilde{\eta}$, and their ratio is given by $\tilde{\eta}/\eta \geq \zeta > 0$. The center of the ellipsoid corresponds to the highlighted anchor point $\mathbf{a}^p$ of the polygon. The outer ellipsoid, shown in red and expanded by a factor of two at the boundary of the maximum inscribed ellipsoid, encloses the polygon. Enclosed within the polygon is a blue inner circle with a radius of $\tilde{\eta}$, and the polygon itself is enclosed by a blue outer circle with a radius of $2\eta$; hence, $B_2(\mathbf{a}^p, \tilde{\eta}) \subseteq p \subseteq B_2(\mathbf{a}^p, 2\eta)$.

$$\zeta^d \eta^d \operatorname{vol} B_d(\mathbf{0}, 1) \leq \tilde{\eta}^d \operatorname{vol} B_d(\mathbf{0}, 1) \leq \operatorname{vol}(p) \leq (d\eta)^d \operatorname{vol} B_d(\mathbf{0}, 1). \tag{47}$$

The first inequality follows the condition $\tilde{\eta}/\eta \geq \zeta$. The second inequality approximates the volume of $p$ from the inside using a ball of radius $\tilde{\eta}$, applying the formula $\operatorname{vol}B_d(\mathbf{0}, r) = r^d \operatorname{vol}B_d(\mathbf{0}, 1)$. The final inequality provides an approximation from the outside, where $d\eta$ represents the radius of an enclosing ball around $p$. Figure 5 offers a schematic illustration of Eq. (47).

Let's consider sub-polytopes formed by cutting $p \in P_L^0$ with a hyperplane passing the anchor point of $p$. These sub-polytopes are children of the root polytope $p$ in a binary branch. This process can be repeated for each sub-polytope, creating a binary tree of depth $k$, with at most $2^k$ leaves, each leaf sub-polytope connected to a path rooted at $p$. We can label the sub-polytope in a path at depth one as $q_1$, the sub-polytope in the path at depth two as $q_2$, and so on, as shown in Figure 6, which provides a schematic description of a path of sub-polytopes in the tree and their hyperplane cuttings. To ensure a child's sub-polytope has a non-empty volume, we assume that all eigenvalues of the largest inscribed ellipsoids of any sub-polytope $q_i$ are greater than zero, and the ratio of the smallest radius to the largest radius of a maximum inscribed ellipsoid is not less than $\zeta$. Let's denote $\eta_k$ and $\tilde{\eta}_k$ as the largest and smallest radii of the principal axes of the maximum volume inscribed ellipsoid of $q_k$. Using Eqs. (46) and (47), we obtain the inequality

$$\zeta^d \eta_k^d \operatorname{vol} B_d(\mathbf{0}, 1) \leq \operatorname{vol}(q_k) \leq (1 - \frac{1}{d})^k \operatorname{vol}(p). \tag{48}$$

As we increase the depth $k$, the volume of $q_k$ decreases to zero. This implies that the sequence $\{\eta_k\}$ decreases to zero as $k$ increases, satisfying the inequality

$$\eta_k \leq \left[ \frac{(1 - \frac{1}{d})^k \operatorname{vol}(p)}{\zeta^d \operatorname{vol} B_d(\mathbf{0}, 1)} \right]^{\frac{1}{d}}. \tag{49}$$

Thus, for each path from root polytope $p$ to leaf sub-polytope $q_k$ of the binary tree, we can find a sufficiently large $k$ such that $(d\eta_k) \leq \frac{\gamma_p}{2}$, where $\gamma_p$ is the radius of the smallest enclosing ball of $p$, and $q_k$ is enclosed by a ball of radius $d\eta_k$. The number of hyperplane cuts required to create the path from $p$ to $q_k$ is equal to $k$, the depth of the path, with each hyperplane cut requiring $d + 1$ parameters.

For a binary tree of depth $k-1$, the number of nodes in the tree is fewer than $2k$. Based on the construction, each node contains one hyperplane with $d+1$ parameters that divides the polytope associated with the node

into two sub-polytopes. Although the number of paths grows exponentially as $2^k$, the number of hyperplane cuts — and hence the parameter count — is determined solely by the number of nodes in the tree, which grows only exponentially as $2k$. Thus, despite the exponential number of paths, few than $2k(d+1)$ parameters are needed to reduce the enclosing radius of $p$ to half its value for all paths in the tree.

Note that each region in $P_l^0$ has a specific depth for its own binary tree. We define $\omega \in \mathbb{N} \cup \{0\}$ as the maximum depth of the binary tree corresponding to any polytope in $P_l^0$ for $l \leq L$ (thus, $\omega \geq k$). Hence, the parameter bound $2k(d+1)$ from the previous argument extends to $2\omega(d+1)$, where $\omega$ serves as the uniform depth bound across all regions in $P_l^0$. Using $\omega$, the number of polytopes in $P_{l+1}^0$ is at most $|P_{l+1}^0| \leq 2^\omega |P_l^0| \leq 2^{l\omega}$ with $|P_0^0| = 1$. Since the subdivision of a polytope requires at most $2\omega(d+1)$ parameters and there are at most $2^\omega$ polytopes, the number of parameters needed to generate $P_{l+1}^0$ from $P_l^0$ is bounded by $2\omega(d+1)2^{l\omega}$.

The number of parameters in $\mathcal{M}_{l+1}^0$, denoted by $\#W_{l+1}$, is equal to the number of parameters in $\mathcal{M}_l^0$, denoted by $\#W_l$, plus the number of parameters required to obtain $P_{l+1}^0$ from $P_l^0$, which is bounded by $2\omega(d+1)2^{l\omega}$. With the base case $\#W_0 \leq 2\omega(d+1)$, we obtain

$$\#W_{l+1} \leq \#W_l + 2\omega(d+1)2^{l\omega} \leq (1+\alpha)\#W_l. \tag{50}$$

Recursively expanding $\#W_l$ in the above inequality for $\omega \in \mathbb{N}$, we obtain

$$\#W_l \leq 2\omega(d+1)\sum_{i=0}^{l-1} 2^{i\omega} = 2\omega(d+1)\frac{2^{l\omega}-1}{2^\omega-1}. \tag{51}$$

Since $1+\alpha \geq \frac{\#W_{l+1}}{\#W_l}$, taking the ratio of consecutive terms gives

$$1+\alpha \geq \frac{2^{(l+1)\omega}-1}{2^{l\omega}-1}. \tag{52}$$

For large $l$, $\frac{2^{(l+1)\omega}-1}{2^{l\omega}-1} \approx 2^\omega$, and hence $\alpha \gtrsim 2^\omega - 1$. Since $\omega \geq 1$, $\alpha \geq 1$ at large $l$.

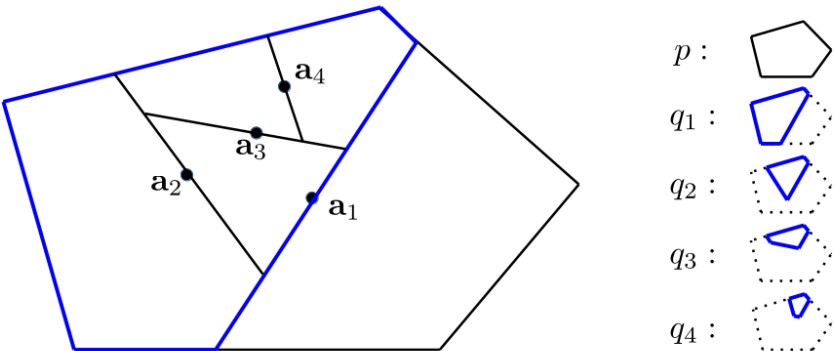

Figure 6: The convex polygon $p$ can be partitioned into sub-polytopes using sequences of hyperplane cuttings, leading to a binary tree structure with $p$ at the root. The root $p$ is divided into two sub-polygons by a hyperplane passing through the anchor point $\mathbf{a}_1$, the center of the maximal inscribed ball of $p$. The sub-polygon $q_1$, colored in blue, is further divided using anchor points $\mathbf{a}_2$, $\mathbf{a}_3$, and $\mathbf{a}_4$, corresponding to the centers of the maximal inscribed balls for sub-polygons $q_2$, $q_3$, and $q_4$, respectively. It is worth noting that the volume of any sub-polygon retains a constant fraction of its parent polygon, as defined by Eq. (46).

To complete the construction, we need to translate all cutting hyperplanes of the trees into a new layer. By induction, suppose we obtain $\mathcal{M}_l^0$. We denote the restriction of $\mathcal{M}_l^0$ on the polytope $p \in \mathcal{P}_l^0$ as $\mathcal{M}_l^0|p$. Let $q \in \mathcal{P}_{l+1}^0$ represent a sub-polytope of $p$. We denote $\mathcal{M}_{l+1}^0|q = \rho\mathbf{A}_{q,p}\mathcal{M}_l^0|p$ the equations define polytope $q$, where $\rho\mathbf{A}_{q,p}$ is the affine mapping derived from the sequence of cutting hyperplanes used to construct $q$ from $p$. In the derivation, we can replace $\rho$ by a diagonal matrix containing entries 0 or 1. The values in this matrix are determined by the values of $\mathcal{M}_l^0|p$. The overall affine linear mapping $M_{l+1}$ is then constructed by stacking the arrays of $\mathbf{A}_{q,p}$ generated from each polytope $p$ in $\mathcal{P}_l^0$.

## D  Proof of Lemma 3

If the center of a ball $b$ with radius $\beta_l$ is located outside the ball $B_{d_e}(\mathbf{x}, \beta_l)$, then the point $\mathbf{x}$ cannot be contained within the ball $b$. We can conclude that the probability of a randomly chosen ball not containing $\mathbf{x}$ is given by $1 - P(\mathbf{x}; B_{d_e}(\mathbf{x}, \beta_l))$, where $P(\mathbf{x}; B_{d_e}(\mathbf{x}, \beta_l))$ represents the probability of $\mathbf{x}$ being located inside the ball $B_{d_e}(\mathbf{x}, \beta_l)$.

The probability of $\mathbf{x}$ being covered by the $l$-th random ball but not by the first $l - 1$ balls by process $X(\beta_l)$ is given by $P(\mathbf{x}; B_{d_e}(\mathbf{x}, \beta_l))(1 - P(\mathbf{x}; B_{d_e}(\mathbf{x}, \beta_l)))^{l-1}$. Hence, the expected number of random balls required to cover $\mathbf{x}$ for the first time is

$$
\begin{aligned}
\bar{l}(\mathbf{x}) &= \sum_{l=1}^{\infty} l P(\mathbf{x}; B_{d_e}(\mathbf{x}, \beta_l))(1 - P(\mathbf{x}; B_{d_e}(\mathbf{x}, \beta_l)))^{l-1} \\
&= P(\mathbf{x}; B_{d_e}(\mathbf{x}, \beta_l)) \frac{\mathrm{d} \sum_{l=1}^{\infty} [1 - P(\mathbf{x}; B_{d_e}(\mathbf{x}, \beta_l))]^l}{\mathrm{d}[1 - P(\mathbf{x}; B_{d_e}(\mathbf{x}, \beta_l))]} \\
&= \frac{1}{P(\mathbf{x}; B_{d_e}(\mathbf{x}, \beta_l))} \leq \frac{1}{c\beta_l^d}.
\end{aligned}
\tag{53}
$$

In the above calculation, we utilized the formula $\sum_{i=0}^{\infty} x^i = \frac{1}{1-x}$ for $x \in (0, 1)$, and $P(\mathbf{x}, B_{d_e}(\mathbf{x}, \beta_l)) \geq c\beta_l^d$ for all $\mathbf{x} \in B_{d_e}(\mathbf{0}, 1)$, which is a consequence of $\mathcal{P} \geq k > 0$ over $B_{d_e}(\mathbf{0}, 1)$.

The mean of the first time to cover the unit $d$-ball by random balls is the integral of the product of $\bar{l}(\mathbf{x})$ and $\mathcal{P}(\mathbf{x})$ over the unit $d$-ball, i.e.,

$$
\mathcal{E}(|X_{min}|) = \int \bar{l}(\mathbf{x}) \mathcal{P}(\mathbf{x}) d\mathbf{x} \leq \frac{1}{c\beta_l^{d_e}}.
\tag{54}
$$

We now estimate the average number of samples drawn independently according to $\mathcal{P}$ and the process $X(\beta_l)$ needed to cover the unit $d_e$-ball for the first time. Leveraging the Markov inequality, we can derive an upper bound for the probability that the initial $N$ samples generated by $X(\beta_l)$ cannot cover the unit $d_e$-ball as follows:

$$
P(|X_{min}| > N) \leq \frac{\mathcal{E}(|X_{min}|)}{N} \leq \frac{1}{cN\beta_l^{d_e}}.
\tag{55}
$$

Hence, the probability of the initial $N$ samples forming a covering is $P(|X_{min}| \leq N) \geq 1 - \frac{1}{cN\beta_l^{d_e}}$.

## E  Proof of Lemma 4

The empirical frequency of bin $p$ in the sequence $S$ is given by

$$
Z_p^m(S) = \frac{1}{N} \sum_{i=1}^{N} \mathcal{I}(X_i \in p) = \frac{n_p(S)}{N},
\tag{56}
$$

where $\mathcal{I}$ is the indicator function and $n_p(S)$ is the number of points in bin $p$. Applying Hoeffding's inequality to $Z_p^m(S)$ yields

$$
P(|\int_p \mathcal{P}(\mathbf{x}) d\mathbf{x} - \frac{n_p(S)}{N}| \geq t_p) \leq 2 \exp(-2Nt_p^2).
\tag{57}
$$

Equivalently,

$$
P(|\int_p \mathcal{P}(\mathbf{x}) d\mathbf{x} - \frac{n_p(S)}{N}| < t_p) > 1 - 2 \exp(-2Nt_p^2),
\tag{58}
$$

where the right-hand side approaches 1 as $N \to \infty$ for any fixed $t_p > 0$. More generally, the same conclusion holds for any sequence $t_p \to 0$ slower than $N \to \infty$. Hence, $\int_p \mathcal{P}(\mathbf{x}) d\mathbf{x} = \frac{n_p(S)}{N}$ in probability as $N \to \infty$.

## F  Generalization bounds via the interpolating zone

When a network $\mathcal{M}$ operates in the interpolating zone where the training error is zero, the histogram-approximation assumption is not required to reach a result analogous to Lemma 2.

**Lemma 5.** *Let $f$ be the function $B_d(\mathbf{0}, 1) \to \{\mathbf{e}_1, \cdots, \mathbf{e}_l\}$, defined as $f = \bigcup_{i=1}^{l} g(i)$, where $g(i) = \{\mathbf{x} | f(\mathbf{x}) = \mathbf{e}_i\}$ is a semi-algebraic set with decision boundary $\partial f$ adhering to the stratification equations (2) and (3). Let $\gamma_s$ and $\beta_l$ represent the smallest enclosing radius and largest inscribed radius of the decision regions in $B_d(\mathbf{0}, 1)$ of the one-hot classification network $\mathcal{M}$. Assume the training data satisfies the conditions for $\beta_l$-covering the unit d-ball and there is no training error. We use the notation $|S|$ to denote the volume of the set $S$.*
*(i) For an arbitrary density function $\mathcal{P}$ on $B_d(\mathbf{0}, 1)$, the error is given by:*

$$\int_{B_d(\mathbf{0},1)} g(f(\mathbf{x}), \mathcal{M}(\mathbf{x}))\mathcal{P}(\mathbf{x}) \, d\mathbf{x} \leq c_\mathcal{P} \sum_{k=0}^{d-1} (2\gamma_s)^{d-k} |B_{d-k}(\mathbf{0}, 1)| \, \|f_k\|, \tag{59}$$

*where $g(f(\mathbf{x}), \mathcal{M}(\mathbf{x}))$ is the 0/1 error, and $c_\mathcal{P} \in (0, 1)$ depends on the density distributed over the $2\gamma_s$-neighborhood of $\partial f$.*
*(ii) We further suppose the volume of $f_{d-1}$ is not zero. For a small value of $\gamma_s$, we can approximate the above expression as follows:*

$$\int_{B_d(\mathbf{0},1)} g(f(\mathbf{x}), \mathcal{M}(\mathbf{x}))\mathcal{P}(\mathbf{x}) \, d\mathbf{x} \leq c_\mathcal{P}(4\gamma_s)|f_{d-1}|. \tag{60}$$

■

**Proof**.

The notation and deduction steps follow those in the proof of Lemma 2. The major deviation is in Eq. (31), in which calculating the error between $f(\mathbf{x})$ and $\mathcal{M}(\mathbf{x})$ with respect to training point $\mathbf{x}_p$, we can use the following equation:

$$
\begin{aligned}
g(f(\mathbf{x}), \mathcal{M}(\mathbf{x})) &= \frac{1}{\sqrt{2}}(\|f(\mathbf{x}) - \mathcal{M}(\mathbf{x}) + f(\mathbf{x}_p) - f(\mathbf{x}_p) + \mathcal{M}(\mathbf{x}_p) - \mathcal{M}(\mathbf{x}_p)\|_2) \\
&\leq \frac{1}{\sqrt{2}}(\|f(\mathbf{x}) - f(\mathbf{x}_p)\|_2 + \|f(\mathbf{x}_p) - \mathcal{M}(\mathbf{x}_p)\|_2 + \|\mathcal{M}(\mathbf{x}_p) - \mathcal{M}(\mathbf{x})\|_2) \\
&\leq g(f(\mathbf{x}), f(\mathbf{x}_p)).
\end{aligned}
\tag{61}
$$

As points in the same decision region of network $\mathcal{M}$ have the same class, the term $\|\mathcal{M}(\mathbf{x}_p) - \mathcal{M}(\mathbf{x})\|_2$ is zero. Meanwhile, $\|f(\mathbf{x}_p) - \mathcal{M}(\mathbf{x}_p)\|_2$ vanishes due to the interpolation condition, where $f(\mathbf{x}_p) = \mathcal{M}(\mathbf{x}_p)$.

Denote $\{\mathbf{x}_{p,i}\}_{i=1}^{n_p}$ as the set of $n_p$ training points in decision region $p$. By applying Eq. (61) to each training point in the region and then taking the maximum, we obtain

$$g(f(\mathbf{x}), \mathcal{M}(\mathbf{x})) \leq \max_i g(f(\mathbf{x}), f(\mathbf{x}_{p,i})). \tag{62}$$

To assess the expected error for a probability distribution $\mathcal{P}$, we apply Eq. (62) to each point $\mathbf{x}$ in region $p$, and then sum over all decision regions to obtain:

$$
\begin{aligned}
\int_{B_d(\mathbf{0},1)} g(f(\mathbf{x}), \mathcal{M}(\mathbf{x}))\mathcal{P}(\mathbf{x}) \, d\mathbf{x} &= \sum_p \int_p g(f(\mathbf{x}), \mathcal{M}(\mathbf{x}))\mathcal{P}(\mathbf{x}) \, d\mathbf{x} \\
&\leq \sum_p \int_p \max_i g(f(\mathbf{x}), f(\mathbf{x}_{p,i}))\mathcal{P}(\mathbf{x})d\mathbf{x} \\
&\leq \sum_p \int_p \max_i \hat{h}_f(\mathbf{x}; \|\mathbf{x} - \mathbf{x}_{p,i}\|_2)\mathcal{P}(\mathbf{x})d\mathbf{x} \\
&\leq \int 1([2\gamma_s \geq \eta_f(\mathbf{x})])\mathcal{P}(\mathbf{x})d\mathbf{x}.
\end{aligned}
\tag{63}
$$

The last inequality arises from the fact that the distance between any two points within the same decision region will always be less than or equal to $2\gamma_s$. Additionally, $\hat{h}_f$ is defined in Eq. (30).

The rest of the derivations follow those of parts (i) and (ii) in the proof of Lemma 2, which we omit. Q.E.D.

This lemma is the principal mechanism underlying the scaling laws with respect to sample size and network size, as given in Theorems 1 and 2. We can thus immediately obtain analogous results by replacing the histogram-approximation assumption with the interpolation condition where the training error is zero. Consequently, the training errors in Eqs. (15) and (16) vanish. To avoid redundancy, we do not restate those results here.

