# OpenReview forum: "The Scaling Laws of Classification Networks: Insights from Adaptive Exact Average Density Approximation"
_TMLR — Rejected by TMLR_

### Review · Reviewer_fPR8 · 2026-03-23

**Summary Of Contributions:**

This paper investigates the theoretical foundations of scaling laws in deep classification networks, with a specific emphasis on the role of network depth and training data size. The authors propose a framework where the generalization error is decomposed into two primary components: 1) error on the training data points and 2) a term governed by the minimum radius ($\gamma_s$) that can cover of the network's decision regions.
The  key aspects are

1) Sample Size Scaling:  The authors argue that the error contribution from sample size depends on the dimension of the data manifold, scaling inversely with the number of training points $N$.

2) Parameter Scaling (Box-Module Dimension): They introduce the "Box-Module Dimension" to quantify a network's efficiency in partitioning the input space. The paper claims that increasing depth allows for a $(1 + \alpha)$ parameter scaling law.

3) Empirical Validation: The findings are presented as a theoretical justification for the power-law behaviors empirically observed in standard architectures, such as ResNet and VGG, on benchmark datasets.

**Additional Comments:**

NA

**Audience:**

Yes

**Audience Explanation:**

This work tackles the important task of formalizing the scaling behavior of deep networks. Its approach and the findings are relevant to theorists interested in generalization bounds and practitioners focused on the empirical performance of standard architectures.

**Broader Impact Concerns:**

I don't see any concerns.

**Claims And Evidence:**

No

**Claims Explanation:**

**Strengths**

1) The paper introduces the Box-Module Dimension, a new construct that attempts to bridge the gap between a network's generalization error and its parameter efficiency. This provides a fresh geometric lens for viewing how deep networks partition high-dimensional space compared to shallow counterparts.

2) The work tackles a highly relevant and challenging problem by attempting to provide a unified explanation for both sample complexity (the role of training data volume) and model capacity (the role of parameter count).

3) By deriving specific scaling exponents and comparing them against the behavior of standard architectures like ResNet and VGG, the paper seeks to provide much-needed theoretical justification for the power-law behaviors observed in large-scale machine learning practice.

**Weakness**
1) The paper ignores the training process (e.g., SGD), and it fails to address whether the configurations mentioned in Proposition 1 are actually reachable via standard optimization.

2) In Eq. 18 and Theorem 1, the authors assume a $\gamma_s$-covering of the data manifold. However, the sample complexity required for such a cover scales as $N = O(1/\gamma_s^d)$. Given that $\gamma_s < 1$, the number of training points $N$ must grow exponentially with the data dimension $d$. For high-dimensional datasets (e.g., ImageNet), the Curse of Dimensionality exists. Also, in Theorem 1, the $\gamma_s$ is upper bounded by a factor that linearly depends on the $\gamma_s$. It's not very informative. This is happening as its actualy bounding $\beta_l$.

3)  In Section B of the Appendix, while deriving the test error bound, the authors approximate the network output $N_s(x)$ using a first-order Taylor expansion of the target function $f(x)$. There is justification provided for why a piecewise-constant or piecewise-linear neural network should satisfy this specific approximation property. Please give some intuition for this approximation

4) In Appendix C, the added parameters at each layer are defined as $(d+1)2^{l\omega}$. This implies that the network width must grow exponentially with depth $l$ to maintain the partition (reducing $\gamma_s$ by 2), a structural property that is not observed in standard architectures like ResNet or VGG. Also, please clarify the induction steps; it's not clear.

5) Are the findings presented in the paper limited to the interpolation regime? since the theory predicts that increasing the model parameters will reduce the test error.

**Inconsistencies and typographical errors**

1) The text below Eq. 43 prints the parameter cost as $2k(d+1)$, but the binary tree logic and the preceding math dictate an exponential cost of $2^k(d+1)$?

2)  The geometric derivation in Eq. 28 appears to be missing a constant scaling factor of $\sqrt{2}$.

3) The current notation in Eq.19 fails to express that the integral is a random variable. Please explicitly mention for better readability or use some informative notation (for example, $p(S)$ instead of p) .  Should the integral be evaluated over the fixed global domain $B_d(0,1)$ ?
4) The authors state that for a ResNet architecture, $\log_2(1 + \alpha) \approx 5$. According to their own derived scaling law, this should result in a 32-fold increase in parameters to achieve the target error reduction. However, the text claims a 10-fold increase is sufficient.

The empirical evidence presented in Tables 1 to 4 is convincing as it supports the theoretical predictions. However, before I could say "yes", I would like the authors to address the mentioned concerns.

**Requested Changes:**

1) Parameter Scaling: Address the contradiction between the exponential parameter requirement $(2^{l\omega})$ and the practical architectures mentioned.

2) Correct Induction Steps: Expand the induction steps in Appendix C; the inequalities are not clear.

3) Justify Eq. 36: Provide a formal justification or intuition for the first-order Taylor approximation of the network output.

4) Address High-Dimensional Covering: Discuss how the $N = O(1/\gamma_s^d)$ requirement relates to real-world high-dimensional datasets.

5) Clarify the bounding of $\gamma_s$ in Theorem 1.

6) Fix Notational and typographical Errors as mentioned above.

---

> ### Author Response · Authors · 2026-04-14
> **The Scaling Laws of Classification Networks: Insights from Adaptive Exact Average Density Approximation**
>
> **Weakness**
>
> 1. ${\bf Response:}$
> Please refer to the highlighted paragraph above the proposition.
>
> 2. ${\bf Response:}$
> We have made modifications to address your concerns. First, we emphasize that the sample complexity is governed by the intrinsic (effective) dimension $d_e$ of the data manifold rather than the ambient dimension $d$, which mitigates the Curse of Dimensionality for datasets such as ImageNet. Second, we have updated the equation and the theorem to be stated in terms of $\beta_l$ rather than $\gamma_s$, making the bound more informative.
>
> 3. ${\bf Response:}$
> We have re-derived the test error bound using the assumptions that the network $\mathcal{N}_s$ is an interpolating network and that each region in the partition of $\mathcal{N}_s$ contains at least one training point. Furthermore, we found a result that does not rely on the assumptions, which we present in the remark of Appendix B.
>
> 4. ${\bf Response:}$
> We have addressed your concern regarding the exponential width growth of our construction and its difference from standard architectures in the highlighted passage surrounding Proposition 1. In Appendix C, the induction steps are clarified in detail.
>
> 5. ${\bf Response:}$
> The findings are not limited to the interpolation regime. In fact, our analysis demonstrates that scaling laws can emerge in two settings: one is the interpolation regime, and the other is when the histogram exactly approximates the average density. Please refer to the remark below Lemma 2 and Appendix F for details.
>
> **Inconsistencies and typographical errors**
>
> 1. ${\bf Response:}$
> The parameter count is determined solely by the operations at the intermediate nodes, not by the number of paths through the tree. Although the number of paths grows exponentially with depth, each region subdivision along any path is determined by the parameters at the intermediate nodes. Since the number of intermediate nodes grows only polynomially, the parameter count remains polynomial. We have added a highlighted paragraph immediately under Eq. (49) to clarify this.
>
> 2. ${\bf Response:}$
> We have corrected the mistake in the updated manuscript.
>
> 3. ${\bf Response:}$
> We clarify that the number of training points ($n_p$) within each region is a random variable in the sentence below Eq. (19) and in Lemma 4.
>
> 4. ${\bf Response:}$
> Thank you for pointing out this mistake. We have fixed it.
>
> **Requested Changes:**
> 1. ${\bf Response:}$
> In our construction, the covering radius is reduced by at least half in each additional layer, which requires doubling the number of partitioning regions at every layer and thus leads to an exponential increase in network width. In contrast, practical architectures such as ResNet and VGG progressively reduce the covering radius across layers, achieving a similar reduction with significantly fewer parameters per layer. Therefore, the exponential parameter requirement is a property of our specific theoretical construction used to establish the existence of scaling laws, and should not be interpreted as a requirement for practical architectures. The distinction between our construction and practical architectures is further explained in the highlighted paragraph immediately above Proposition 1.
>
> 2. ${\bf Response:}$
> We have expanded the induction steps in Appendix C. Please refer to the highlighted paragraphs, particularly Eqs. (50)-(52), for details.
>
> 3. ${\bf Response:}$
> Eq. (36) in the previous manuscript corresponds to Eq. (42) in the revised manuscript. We have removed the Taylor expansion by instead imposing that the network interpolates the training points and that each hypercube in the partition of the input domain contains at least one training point. Additionally, we present an alternative bound on the test error, which does not require these assumptions. Please refer to the highlighted parts in Appendix B for details.
>
> 4. ${\bf Response:}$
> We have emphasized the role of the effective (intrinsic) dimension in mitigating the Curse of Dimensionality. Please refer to the highlighted parts in the Assumptions and Simplifications section and the first paragraph of Section 5. Additionally, we have modified Theorem 1 and Lemma 3, with $\gamma_s$ replaced by $\beta_l$ and $d$ replaced by $d_e$, the effective (intrinsic) dimension of the data manifold. The significance of sample size reduction when considering $d_e$ can also be found in Eq. (27) of the revised manuscript.
>
> 5. ${\bf Response:}$
> We have replaced $\gamma_s$ by $\beta_l$ using the fact that $\beta_l \geq \gamma_s$, since $\beta_l$ is the circumradius and $\gamma_s$ is the inradius of the polytopes. This substitution removes $\gamma_s$ from the theorem and expresses the bound directly in terms of the covering radius $\beta_l$, making it more interpretable.
>
>
> 6. ${\bf Response:}$
> Thank you for your careful reading and for pointing out these critical issues.

---

> > ### Comment · Reviewer_fPR8 · 2026-04-22
> >
> > I thank the authors for their constructive revision; most of my concerns are addressed. However, one final concern is regarding the updated proof of Theorem 1. At the conclusion of Theorem 1, the text states: "Applying this inequality and $\beta_l \ge \gamma_s$ (since $\beta_l$ is the circumradius and $\gamma_s$ is the inradius)." This contradicts the geometric definitions in Section 3.2. Geometrically, an enclosing ball (circumradius) must be larger than an inscribed ball (inradius). Thus, it must be $\gamma_s \ge \beta_l$. Swapping these definitions invalidates the final substitution.
> > However, since these quantities are finite and non-vanishing, one can argue that $\gamma_s \le K \beta_l$ for some K > 1.

---

> ### Author Response · Authors · 2026-04-24
> **Revision of The Scaling Laws of Classification Networks: Insights from Adaptive Exact Average Density Approximation**
>
> Comment:
> I thank the authors for their constructive revision; most of my concerns are addressed. However, one final concern is regarding the updated proof of Theorem 1. At the conclusion of Theorem 1, the text states: "Applying this inequality and
>  (since
>  is the circumradius and
>  is the inradius)." This contradicts the geometric definitions in Section 3.2. Geometrically, an enclosing ball (circumradius) must be larger than an inscribed ball (inradius). Thus, it must be
> . Swapping these definitions invalidates the final substitution. However, since these quantities are finite and non-vanishing, one can argue that
>  for some K > 1.
>
>  ${\bf Response:}$
>
>  Thank you for catching this error. You are correct that the circumradius must satisfy $\gamma_s \geq \beta_l$ (circumradius ≥ inradius), and our statement in the original proof had these definitions inverted. We have corrected the proof of Theorem 1 accordingly: rather than claiming $\beta_l \geq \gamma_s$, we now write $K \beta_l \geq \gamma_s$ for some constant $K \geq 1$, which follows from the fact that both quantities are finite and non-vanishing. The corrected passage is highlighted in purple in the revised manuscript.
> We have also made a corresponding correction to the Remark in Section 5, also highlighted in purple, to ensure consistency throughout the paper.

---

### Review · Reviewer_ZVQs · 2026-03-28

**Summary Of Contributions:**

This paper involves analyzing the scaling laws for deep neural networks.
In particular, the paper provides a strong theoretical contribution by linking the geometric properties of a network's decision boundaries to empirical scaling laws.
The authors introduce the concept of "box-module dimension" to quantify how many parameters are needed to describe a shape at different scales is a clever adaptation of fractal geometry (box-counting dimension) to neural network architecture.
The authors successfully decouple the scaling laws, showing that sample size scaling is intrinsically linked to the data manifold's effective dimension, while network size scaling is tied to the learning method and architecture.
Lastly, the authors provides some empirical evidence to support their theoretical claims.

**Audience:**

Yes

**Audience Explanation:**

This paper deals with the analysis of scaling laws for deep classification networks. The presented results could be helpful in designing the deep neural networks.

**Broader Impact Concerns:**

N.A.

**Claims And Evidence:**

Yes

**Claims Explanation:**

The claims in this paper are sound and well-supported by theoretical analysis.
There are only a few questions and concerns, which I will list in the "Requested Changes" section below.

**Requested Changes:**

* Top of page 2, duplicate "The The"
* Some math symbols are not rendered correctly, e.g., $\underline{0}$ in lemma 2, which should not be italicized.
* page 6, "ruggedness", I suppose the wrong quote command is used in the LaTeX source, which should be ` ``ruggedness'' ` instead of ` "ruggedness" `.
* The structure of the experiments from page 9 to page 11 is not quite aligned with the previous section titles.
For example, this sentence on page 9: "Findings on ResNet and VGG models for ImageNet and CIFAR-100" should be a title, but appears as a sentence without punctuation.
The subsections I., II., and III. of this part look like they should be subsections of under the previous title.
It is recommended to restructure this part of the paper, e.g., moving the whole experiments section to a new subsection 4.2.3, so that the structure of the paper is more clear and consistent.
* Bottom of page 9: title "II. ImageNet on CIFAR-100", do you mean "II. ResNets on CIFAR-100"?

Besides, if the authors could address the following questions, this paper would be substantially strengthened, but these will not substantially affect the decision on this paper:

* One of the most fundamental assumptions in this paper is that the histogram of sample points accurately reflects the regional average density.
This assumption can be fulfilled when the sample size is sufficiently large, e.g., $N \to \infty$.
However, in practice, the sample size is always finite, and it is not clear how large the sample size needs to be for this assumption to hold.
A brief discussion on how this assumption holds up in lower-data regimes would be helpful.
* In the experiments section, drawing a line through just two points to extrapolate the performance of models that are magnitudes larger is mathematically fragile.
While the resulting relative errors are impressively low, achieving this on a handful of datasets using 2-point regressions feels more like a preliminary proof-of-concept than a solid empirical evidence.
It would be very helpful if the authors could conduct more experiments on a much wider variety of architectures, datasets, and a more robust regression methodology.

---

> ### Author Response · Authors · 2026-04-14
> **The Scaling Laws of Classification Networks: Insights from Adaptive Exact Average Density Approximation**
>
> **Are the claims made in the submission supported by accurate, convincing and clear evidence?:** Yes
> Explain your answer above:
> The claims in this paper are sound and well-supported by theoretical analysis. There are only a few questions and concerns, which I will list in the "Requested Changes" section below.
>
> **Would at least some individuals in TMLR's audience be interested in knowing the findings of this paper?:** Yes
> Explain your answer above:
> This paper deals with the analysis of scaling laws for deep classification networks. The presented results could be helpful in designing the deep neural networks.
>
> **Requested Changes:**
>
> $\bullet$ Top of page 2, duplicate "The The"
> Some math symbols are not rendered correctly, e.g.,
>
> $\bullet$ in lemma 2, which should not be italicized.
>
> $\bullet$ page 6, "ruggedness", I suppose the wrong quote command is used in the LaTeX source, which should be ``ruggedness'' instead of "ruggedness".
>
> ${\bf Response:}$ The above three typos are fixed.
>
> $\bullet$ The structure of the experiments from page 9 to page 11 is not quite aligned with the previous section titles. For example, this sentence on page 9: "Findings on ResNet and VGG models for ImageNet and CIFAR-100" should be a title, but appears as a sentence without punctuation. The subsections I., II., and III. of this part look like they should be subsections of under the previous title. It is recommended to restructure this part of the paper, e.g., moving the whole experiments section to a new subsection 4.2.3, so that the structure of the paper is more clear and consistent.
> Bottom of page 9: title "II. ImageNet on CIFAR-100", do you mean "II. ResNets on CIFAR-100"?
>
> ${\bf Response:}$ We have restructured the section accordingly. A new section 4.2.3 is introduced to categorize the different experiments into distinct parts.
>
> $\bullet$ Besides, if the authors could address the following questions, this paper would be substantially strengthened, but these will not substantially affect the decision on this paper:
>
> One of the most fundamental assumptions in this paper is that the histogram of sample points accurately reflects the regional average density. This assumption can be fulfilled when the sample size is sufficiently large. However, in practice, the sample size is always finite, and it is not clear how large the sample size needs to be for this assumption to hold. A brief discussion on how this assumption holds up in lower-data regimes would be helpful.
>
> ${\bf Response:}$
>
> We have conducted a preliminary analysis to estimate the required order of sample sizes in response to this comment.  Please see the remark in Section 5 of the revised manuscript, where we provide a partial estimate of the order of magnitude for the number of training points under the conditions that the training points form a $\beta_l$-covering of the input domain and that each decision region contains at least one training point. Additionally,  in  Eq. (27), there is an interesting parameter interplay between the effective dimension and box-modulus dimension, captured by $d_e/\log_2(1+\alpha)$, which  characterizes the network size.
>
> $\bullet$ In the experiments section, drawing a line through just two points to extrapolate the performance of models that are magnitudes larger is mathematically fragile. While the resulting relative errors are impressively low, achieving this on a handful of datasets using 2-point regressions feels more like a preliminary proof-of-concept than a solid empirical evidence. It would be very helpful if the authors could conduct more experiments on a much wider variety of architectures, datasets, and a more robust regression methodology.
>
>
> ${\bf Response:}$
>
> In response to your comment, we have included the Udacity dataset and the InceptionNet family in Table 5. The revised manuscript contains two prediction methods, with results presented in Tables 1, 2, and 5. However, due to the limitation of only three data points reported in Tables 3 and 4, the slope estimation is based on the two smallest models. We agree that the proposed methods are prototypes, and their adaptation to practical settings requires more experiments and further investigation.

---

> ### Author Response · Authors · 2026-04-24
> **Revision of the scaling laws of classification networks: Insights from adaptive exact average density approximation**
>
> We have uploaded a revised manuscript in which the proof of Theorem 1 and the Remark in Section 5 have been corrected. The relevant changes are highlighted in purple for ease of review. We thank the reviewer for bringing this to our attention.

---

### Review · Reviewer_JXWu · 2026-03-30

**Summary Of Contributions:**

The paper investigates scaling laws in neural networks, both theoretically (by expressing the complexity of the decision boundary in terms of upper/ lower bounds of "balls") and practically, by providing an empirical study on 3 tasks. With the focus on the complexity of the decision boundary (expressed as box-module dimension), authors focus on studying models of different depths. The novelty is that the slope of the scaling law is determined by the complexity analysis, rather than by studying multiple existing datapoints and extrapolating.

Strengths

The proposed approach (deriving the scaling behavior in a principled way from the data and model, rather than empirically) is an interesting approach.

Weaknesses

The paper is written confusingly and mixes novel contributions and known relationships (see below). I have a hard time teasing out the exact contribution of the paper.

**Additional Comments:**

none

**Audience:**

Yes

**Audience Explanation:**

The paper treats an important and relevant question in the literature, it would be of interest to many readers.

**Claims And Evidence:**

No

**Claims Explanation:**

- since the slope can be computed, rather than measured, can it be applied to other starting points? i.e. (for the argument's sake) predict performance of smaller models from bigger models? if so, what are the results?
- most papers study an even larger set of tasks (Udacity, CIFAR10, etc), would it be possible to extend the analysis to these tasks and maybe a larger set of models as well?
- what is the impact of the "label error" (of various datasets) on the presented results?

**Requested Changes:**

Please provide a clear thesis of the paper's original contributions and the relevance of the novelty. I am not clear how the "generalization bounds" have been set in this paper and how it was shown that they align with empirical scaling law in ways that go beyond established literature. Also, how has prior work "not placed enough emphasis on network depth" since it covers networks of all shapes and forms?

Addressing the questions in the claims section would also be required for me to recommend acceptance of this paper.

---

> ### Author Response · Authors · 2026-04-14
> **The Scaling Laws of Classification Networks: Insights from Adaptive Exact Average Density Approximation**
>
> **Weaknesses**
> The paper is written confusingly and mixes novel contributions and known relationships (see below)...
>
> ${\bf Response:}$
> Explaining the scaling-law behavior observed empirically in deep neural networks is a widely studied and active subject. The laws are studied in the literature along two distinct directions --- one characterizing scaling with respect to sample size and the other with respect to the number of parameters --- and no single framework has been shown to unify both. We have reviewed relevant literature on sample-size direction and conducted a thorough comparison with the outstanding work of Sharma and Kaplan (2022) on the parameter-scaling direction in  Appendix B.
>
> Our contributions are listed as follows:
> $\bullet$ Decoupling of scaling laws: We show that sample-size scaling is intrinsically linked to the data manifold's effective dimension, while network-size scaling is governed by the learning method and architecture. This decoupling clarifies the distinct roles each component plays in the emergence of scaling behavior.
> $\bullet$ Box-module dimension:
> We introduce the concept of box-module dimension as a formal measure of the rate at which the number of parameters required to describe a geometric shape grows across scales - providing a scale-sensitive complexity measure suited to the analysis of neural network representations.
> $\bullet$ Scaling laws beyond interpolation.
>
> We have incorporated these contributions into the Introduction section of the revised manuscript.
>
> **Are the claims made $\cdots$ Explain your answer above:**
>
> $\bullet$ since the slope can be computed, rather than measured, can it be applied to other starting points? i.e. (for the argument's sake) predict performance of smaller models from bigger models? if so, what are the results?
>
> ${\bf Response:}$
> As you indicated, the robustness of the prediction depends on how one selects the points used to calculate the slope. The results we report use the points corresponding to the two smallest models reported in each paper to derive the slope. We do not consider selecting different sets of points to calculate this value, as per your suggestion, mainly because, in some datasets and models (e.g., Tables 3 and 4), there are only three measured points. This limits our selection of different sets of points for slope estimation. Nevertheless, to address this concern, we have included an additional method to predict the test errors across datasets and models. Please see the prediction results in Tables 1, 2, and 5 in the revised manuscript and the description of the method in the highlighted part of Section 4.2.3.
>
> $\bullet$ most papers study an even larger set of tasks (Udacity, CIFAR10, etc), would it be possible to extend the analysis to these tasks and maybe a larger set of models as well?
>
> ${\bf Response:}$
> ImageNet and CIFAR-100 have more images than CIFAR-10 and Udacity. In accordance with your suggestion, we present the prediction results for the Udacity dataset, as shown in Table 5.
>
> $\bullet$ what is the impact of the "label error" (of various datasets) on the presented results?
>
> ${\bf Response:}$ The label errors in the ImageNet and CIFAR-100 datasets are given in the captions of Tables 1 and 2, respectively.
>
> If a dataset is contaminated with labeling noise, the underlying density distribution will differ from the true one. In that case, the effective dimension increases and, moreover, it becomes harder to meet our assumption that the histogram approximates the underlying density function, and it is also harder to derive the interpolating network. Beyond these considerations, our analysis results appear to remain applicable. However, this subject warrants further investigation to avoid overclaiming.
>
> **Requested Changes:**
>
> ${\bf Response:}$
> The contribution and novelty are rewritten and highlighted in the Introduction of the revised manuscript.
>
> The scaling law indicates that the generalization error bound, which is the difference between the expected test error and the expected training error, behaves as a line in the log-log plot of generalization error and the number of parameters, as shown in Figure 2. Compared to Eq. (16), the generalization error bound is inversely proportional to the number of parameters with exponent $\log_2(1+\alpha_{\mathcal M})$. In the log-log plane of generalization error versus the number of parameters, this formula becomes a line with slope $-1/\log_2(1+\alpha_{\mathcal M})$. We have explained and highlighted it Part I in Section 4.2.3 of the revised manuscript.
>
> The remark that prior work does not place enough emphasis on network depth serves two purposes: to contrast our analysis of the scaling law with respect to results in the literature that concern shallow networks only (particularly, in Appendix B); and to highlight the effectiveness of using the box-module dimension in characterizing the scaling law for networks of varying depths and architectures.

---

> ### Author Response · Authors · 2026-04-24
> **Revision of the scaling laws  of classification networks: Insights from adaptive exact average density approximation**
>
> We have uploaded a revised manuscript in which the proof of Theorem 1 and the Remark in Section 5 have been corrected. The relevant changes are highlighted in purple for ease of review. We thank the reviewer for bringing this to our attention.

---

### Decision · Action_Editor_Pai3 · 2026-05-04

**Recommendation:** Reject

**Additional Comments:**

Actually, I would suggest a major revision (there is no such option for me to choose as an action editor) to focus on making claims more conservative. The authors should explicitly state which conclusions are theoretical bounds under assumptions, which are empirical observations, and which are hypotheses requiring further validation. They should also address Reviewer JXWu’s concern by clarifying the effect of label error and avoiding unsupported claims about applicability under label noise.

**Audience:**

Yes

**Audience Explanation:**

Scaling laws for classification networks are of clear interest to TMLR readers. The paper offers a geometric perspective, including effective dimension and box-module dimension, that may interest theorists studying generalization and practitioners interested in predicting performance across model scales.

**Claims And Evidence:**

No

**Claims Explanation:**

The core theoretical results are sound under the stated assumptions, and the empirical results provide preliminary support. However, the paper should revise its tone to clearly distinguish what is proved from what is empirically illustrated, and avoid suggesting broader validity beyond the assumptions or experiments, especially regarding label noise and practical optimization.

**Resubmission Of Major Revision:**

The authors may consider submitting a major revision at a later time.